# Differentially Private Submodular Maximization with a Knapsack Constraint

**Ron Zadicario** [1]    **Tova Milo** [1]

## Abstract

Submodular maximization subject to a knapsack constraint (SMK) is a fundamental problem in discrete optimization, with wide-ranging applications in machine learning and related fields. As these applications increasingly involve sensitive individual data, there is a growing need for high-utility algorithms that provide formal privacy guarantees. In this work, we study the SMK problem under *differential privacy*, considering both monotone and non-monotone objective functions. For monotone objectives, we propose a differentially private algorithm that achieves the optimal $(1 - 1/e)$-approximation ratio while significantly improving both additive error and query complexity over prior work. We also present a more efficient algorithm for the same setting, achieving a $1/2$-approximation. For non-monotone objectives, we introduce, to our knowledge, the first differentially private algorithm with provable guarantees, achieving a $1/4$-approximation in expectation and an additive error comparable to the best known for monotone objective functions.

## 1. Introduction

Constrained maximization of submodular functions is a cornerstone of discrete optimization. Submodularity formalizes the notion of *diminishing returns*: an element's marginal contribution to a set decreases as the set grows. This property is ubiquitous in theory and practice, driving widespread interest in submodular maximization across machine learning (ML), data science, and related fields. As the problem is NP-hard in general, significant effort has been devoted to designing efficient approximation algorithms. These have been applied to a variety of tasks, including feature selection (Krause & Guestrin, 2005), personalized recommendation (Mirzasoleiman et al., 2016), influence

maximization (Kempe et al., 2003), exemplar-based clustering (Gomes & Krause, 2010), information gathering (Krause et al., 2008), active learning (Esfandiari et al., 2021), data subset selection (Wei et al., 2015), and interpretable rule learning (Lakkaraju et al., 2016). Many of these applications involve non-monotone objectives due to diversification or regularization penalties.

One of the earliest and most studied problem variants involves elements with associated costs, and feasible solutions are constrained by a budget. This constitutes the classical problem of submodular maximization subject to a knapsack constraint (SMK), which has attracted sustained attention for over four decades (Wolsey, 1982). More recently, submodular maximization has been applied in settings involving sensitive individual data, where it is crucial that the optimization procedure simultaneously ensures rigorous privacy guarantees alongside strong utility. As an illustration, consider the following feature selection problem

*Example* 1.1. A sensitive medical dataset $\{(\mathbf{x}_i, y_i)\}_{i=1}^n$ associates each individual $i$ with a feature vector $\mathbf{x}_i = (\mathbf{x}_i(1), \ldots, \mathbf{x}_i(m))$ and a binary label $y_i$ indicating the presence of a disease. Each feature $j \in [m]$ (e.g., blood test result) incurs an acquisition cost $c_j$. The goal is to select a subset of features $S \subseteq [m]$ that enables accurate prediction of $y$ while satisfying the budget constraint $\sum_{j \in S} c_j \leq B$. Following Krause & Guestrin (2005); Mitrovic et al. (2017), a natural approach models this task as maximizing a submodular objective that captures the mutual information between the selected features and the label. Crucially, the feature selection process must preserve the privacy of all individuals contributing to the dataset.

In this work, we study the monotone and non-monotone SMK problem under *differential privacy* (Dwork et al., 2006), a rigorous notion that has become the standard for statistical analysis of sensitive data under strong privacy guarantees. Intuitively, the output distribution of a differentially private (DP) algorithm changes only slightly when a single individual's data is modified, thereby ensuring that individual-level information is not disclosed. DP submodular maximization has attracted increasing attention, most commonly in settings where the ground set $\mathcal{N}$ is public, while the submodular objective $f_D : 2^{\mathcal{N}} \to \mathbb{R}$ depends on a sensitive dataset $D$ in a problem-specific manner. The goal is to privately find a subset $S \subseteq \mathcal{N}$ that approximately maxi-

[1]Tel Aviv University, Israel. Correspondence to: Ron Zadicario <ronzadicario@mail.tau.ac.il>.

*Proceedings of the 43rd International Conference on Machine Learning*, Seoul, South Korea. PMLR 306, 2026. Copyright 2026 by the author(s).

mizes $f_D$ subject to given feasibility constraints. Most prior work has focused on cardinality constraints (Gupta et al., 2010; Mitrovic et al., 2017) or matroid constraints (Rafiey & Yoshida, 2020; Chaturvedi et al., 2021), while knapsack constraints have, to our knowledge, only been addressed by Sadeghi & Fazel (2021). However, their work leaves room for improvement in several important aspects.

First, their approach relies on exact evaluation of the *multilinear extension* of the submodular function, which may require as many as $2^{|\mathcal{N}|}$ value oracle queries to $f_D$[1]. Second, it is restricted to monotone functions, thereby excluding a variety of important applications. Third, the additive error in their utility guarantee exhibits a suboptimal dependence on both the ground set size and the element costs (see Table 1 for a comparison with our results).

Motivated by these limitations, we propose DP algorithms for monotone and non-monotone SMK that bridge existing gaps in utility and query complexity. The privatization of combinatorial algorithms for SMK gives rise to challenges absent in matroid or cardinality constraint settings, requiring different techniques. These challenges, along with the formal statements of our results, are detailed in Section 4. Our contributions are summarized as follows:

**Monotone.** We present a DP algorithm for monotone SMK that *(i)* achieves the optimal $(1 - 1/e)$-approximation, *(ii)* has a query complexity comparable to that of the currently most practical non-private $(1 - 1/e)$-approximation algorithm, and in particular polynomial for any submodular objective, *(iii)* achieves an additive error that improves upon prior work in two ways: it has polylogarithmic dependence on the ground set size, rather than polynomial, and scales with the maximum size of a feasible solution, rather than the minimum element cost. Row (i) of Table 1 summarizes this result. We further introduce a faster algorithm achieving $1/2$-approximation with improved additive error, as shown in row (ii) of Table 1.

**Non-monotone.** We present the first DP algorithm for non-monotone SMK with provable guarantees. Our algorithm achieves a $1/4$-approximation in expectation, matching the state-of-the-art among combinatorial non-private algorithms. Furthermore, it incurs an additive error comparable to that of the monotone case while maintaining a practical query complexity. This result is summarized in row (iii) of Table 1.

---

[1]While the multilinear extension can be approximated via sampling in polynomial time, doing so would require a separate analysis beyond that in (Sadeghi & Fazel, 2021). Even with such approximations, combinatorial algorithms are typically faster than continuous relaxation-based approaches (Feldman et al., 2023).

## 2. Related Work

In this section, we survey related work on non-private SMK, DP submodular maximization, and lower bounds. Throughout, $k$ denotes the maximum size of a feasible solution, and $n$ the ground set size.

**Non-Private Monotone SMK.** For cardinality-constrained submodular maximization, the classical greedy algorithm achieves a tight $(1 - 1/e)$-approximation (Nemhauser et al., 1978). Its SMK extension, Density-Greedy (Wolsey, 1982), selects elements of largest *density*, i.e., the ratio between marginal contribution and cost. While it lacks a constant-factor approximation (Feldman et al., 2009), several variants obtain provable but suboptimal guarantees (Wolsey, 1982; Khuller et al., 1999). Sviridenko (2004) first achieved the optimal $(1 - 1/e)$-approximation by guessing three elements from the optimal solution and running Density-Greedy on the residual instance. This was later improved to two elements by Feldman et al. (2023). The resulting `Two-Guess-Greedy` algorithm remains the most practical tight-approximation method, yet still requires enumerating all $O(n^2)$ pairs. Recent works have obtained a $(1/2 - \varepsilon)$-approximation with near-linear query complexity (Yaroslavtsev et al., 2020; Han et al., 2021), while Li et al. (2022) achieved the same guarantee using $O_\varepsilon(n)$ queries.

**Non-Private Non-Monotone SMK.** The best-known approximation for non-monotone SMK is 0.385, obtained by combining the continuous optimization framework of Buchbinder & Feldman (2019) with the rounding scheme of Kulik et al. (2013). This approach requires computationally expensive operations such as optimizing the multilinear extension, which motivates faster combinatorial algorithms. Mirzasoleiman et al. (2016) obtained a deterministic $1/10$-approximation, followed by a deterministic $1/6$-approximation by Han et al. (2021). Amanatidis et al. (2020) introduced a randomized density-greedy variant where each selected element is discarded with some probability, achieving a $1/(3 + 2\sqrt{2})$-approximation in expectation. This was later improved by Han et al. (2021) to $1/4$, which remains the best practical guarantee.

**Differentially Private Submodular Maximization.** DP submodular maximization has been studied in two main settings. The first line of work focuses on $\Delta$-*decomposable* objective functions (Gupta et al., 2010; Chaturvedi et al., 2021; 2023; Ghazi et al., 2024). These take the form $f(S) = \sum_x f_x(S)$, where each individual $x$ contributes a submodular function $f_x : 2^{\mathcal{N}} \to [0, \Delta]$. However, many applications involve submodular functions with more general dependence on the sensitive dataset (e.g., the objective in Example 1.1). A second line of work therefore considers the more general setting of functions with *bounded sensitivity* $\Delta$, which we adopt in this work.

*Table 1.* Guarantees of $(\varepsilon, \delta)$-DP algorithms for SMK with a $\Delta$-sensitive objective function. Constants are omitted. Here $n$ denotes the size of the ground set, $k$ denotes the maximum cardinality of a feasible solution, $L = \max_u f(u)$, and $\beta$ is the failure probability. The budget is normalized to $B = 1$, which implies $1/c_{\min} \geq k$. [†] This bound is derived from Lemma 4 in (Sadeghi & Fazel, 2021). [‡] Their analysis relies on exact evaluation of the multilinear extension, which, in the value-oracle model, requires $2^n$ queries in general.

| Monotone | Approximation | Additive Error | Query Complexity | Algorithm | |
|---|---|---|---|---|---|
| Yes | $1 - 1/e$ | $\frac{\Delta k^{1.5}}{\varepsilon} \sqrt{\log \frac{n}{\delta \beta}} \log \frac{n}{\beta}$ | $O(\beta^{-1} n^3 k)$ | Algorithm 2 **(Ours)** | (i) |
| Yes | $1/2$ | $\frac{\Delta k^{1.5}}{\varepsilon} \sqrt{\log \frac{1}{\delta}} \log \frac{n}{\beta}$ | $O(nk)$ | Algorithm 7 **(Ours)** | (ii) |
| No | $1/4$ (Expected) | $\frac{\Delta k}{\varepsilon} \sqrt{(k + \log \frac{1}{\delta}) \log \frac{1}{\delta}} \log \frac{n}{\beta}$ | $O(nk)$ | Algorithm 3 **(Ours)** | (iii) |
| Yes | $1 - 1/e$ | $\frac{\Delta}{c_{\min} \varepsilon} \sqrt{\frac{1}{\beta} \log \frac{1}{\delta}} \, n \log \frac{n}{\beta} + \frac{\beta L}{c_{\min}^2}$ [†] | $O(2^n)$ [‡] | (Sadeghi & Fazel, 2021) | (iv) |

As a simple baseline, applying the exponential mechanism (McSherry & Talwar, 2007) to select among all feasible subsets yields a 1-approximation with additive error $O(\Delta k \log n/\varepsilon)$. However, it incurs an exponential query complexity. Mitrovic et al. (2017) proposed efficient algorithms for monotone objectives under cardinality constraints, achieving a $(1 - 1/e)$-approximation with $\tilde{O}(\Delta k^{1.5}/\varepsilon)$ additive error.[2] For non-monotone objectives, they obtained a $(1 - 1/e)/e$-approximation in expectation with the same error, and further extended their results to $p$-extendable system constraints. Other notable works include Rafiey & Yoshida (2020) on matroid constraints, Sadeghi & Fazel (2021) on bounded-curvature objectives, and Chaturvedi et al. (2023) on streaming algorithms. In contrast, SMK has been addressed only by the aforementioned work of Sadeghi & Fazel (2021), who among other results, gave an $(\varepsilon, \delta)$-DP algorithm for monotone functions achieving the guarantees stated in row (iv) of Table 1.

**Lower Bounds.** Existing lower bounds on the expected additive error for DP submodular maximization under a cardinality constraint $k$ are $\Omega(k \log(n/k)/\varepsilon)$ for $\varepsilon$-DP algorithms (Gupta et al., 2010), and $\Omega(kc \log(\varepsilon/\delta)/\varepsilon)$ for $(\varepsilon, \delta)$-DP, $c$-approximation algorithms (assuming $n \geq k(e^\varepsilon - 1)/\delta$ and $c \geq 4\delta/(e^\varepsilon - 1)$) (Chaturvedi et al., 2023). As cardinality is a special case of a knapsack constraint, these bounds extend to our setting. While our additive error exceeds them by $\tilde{O}(\sqrt{k})$, these lower bounds are derived from 1-decomposable objectives. In contrast, we consider the broader class of bounded-sensitivity submodular functions, and the dependence on $k$ in our bounds matches the best known for polynomial-time algorithms in the simpler cardinality-constrained setting.

## 3. Preliminaries

In this section, we introduce notation, submodular functions, and key differential privacy concepts used throughout the

---

[2] $\tilde{O}$ hides polylogarithmic factors in the problem parameters.

paper. Additional details on DP generalized selection and omitted pseudo-codes are provided in Appendix A.

**Differential Privacy.** A dataset is a tuple $D = (x_1, \ldots, x_m) \in \mathcal{X}^m$ of items from domain $\mathcal{X}$ where $x_i$ is individual $i$'s data. Two datasets $D$ and $D'$ are called *neighboring* ($D \sim D'$) if they differ in at most one entry. Differential privacy ensures that the output distribution of an algorithm changes minimally when one record is modified:

**Definition 3.1** (Differential Privacy, (Dwork, 2006))**.** A randomized algorithm $\mathcal{A}$ is $(\varepsilon, \delta)$-differentially private (DP) if for any $D \sim D'$ and $S \subseteq \mathrm{Range}(\mathcal{A})$,

$$\mathbf{Pr}[\mathcal{A}(D) \in S] \leq e^\varepsilon \mathbf{Pr}[\mathcal{A}(D') \in S] + \delta.$$

If $\delta = 0$, we say $\mathcal{A}$ is $\varepsilon$-DP.

Our privacy analysis uses the following standard composition results.

**Theorem 3.2** (Dwork et al. (2010))**.** *Let $\mathcal{A}_1, \ldots, \mathcal{A}_k$ be $\varepsilon_0$-DP algorithms. Their $k$-fold adaptive composition $\mathcal{A}_{[k]}$, which outputs $y_i = \mathcal{A}_i(D, y_1, \ldots, y_{i-1})$ for $i = 1, \ldots, k$, is: (i) $k\varepsilon_0$-DP (Basic Composition), and (ii)($\sqrt{2k \log(1/\delta)}\varepsilon_0 + k\varepsilon_0(e^{\varepsilon_0} - 1), \delta$)-DP for any $\delta > 0$ (Advanced composition). To achieve $(\varepsilon, \delta)$-DP for $\varepsilon \leq 1$, it suffices that each $\mathcal{A}_i$ is $\frac{\varepsilon}{2\sqrt{2k \log(1/\delta)}}$-DP.*

The *sensitivity* of a function $q : \mathcal{X}^m \to \mathbb{R}$ measures the maximum change in its value across neighboring datasets:

$$\Delta_q = \max_{D \sim D'} |q(D) - q(D')|.$$

A canonical DP mechanism for privately answering numerical queries is the Laplace mechanism:

**Theorem 3.3** (Laplace mechanism (Dwork et al., 2014))**.** *Suppose $q : \mathcal{X}^m \to \mathbb{R}$ has sensitivity $\Delta_q$. The mechanism returning $q(D) + Y$ with $Y \sim \mathrm{Lap}(\Delta_q/\varepsilon)$ is $\varepsilon$-DP.*

**Private Selection.** Selecting an element $u$ from a finite set $\mathcal{C}$ to approximately maximize a quality score $q(u; D)$

is a fundamental task in private data analysis. The *Report Noisy Max* mechanism with exponential noise[3] (Dwork et al., 2014), denoted $\mathtt{RNM}_\varepsilon(\mathcal{C}, q; D)$, is a standard approach for solving this task accurately and efficiently. This mechanism outputs $u^* \in \arg\max_{u \in \mathcal{C}}\{q(u; D) + Y_u\}$, where $Y_u \sim \mathrm{Exp}(\frac{\varepsilon}{2\Delta})$ and $\Delta = \max_{u \in \mathcal{C}} \Delta_{q(u; \cdot)}$ is the maximal sensitivity of $q$ across all candidates.

**Theorem 3.4** (Dwork et al., 2014). $\mathtt{RNM}_\varepsilon(\mathcal{C}, q; D)$ *is $\varepsilon$-DP. Moreover, with probability $1 - \beta$, the output $u^*$ satisfies* $q(u^*; D) \geq \max_{u \in \mathcal{C}} q(u; D) - (2\Delta/\varepsilon) \log(|\mathcal{C}|/\beta)$.

The error of $\mathtt{RNM}$ scales with the maximal candidate sensitivity, which is insufficient in our setting (see Section 4). To achieve improved utility, we leverage the generalized mechanism of Raskhodnikova & Smith (2016), denoted $\mathtt{GRNM}_\varepsilon(\mathcal{C}, q, \beta; D)$. Given a failure probability $\beta$, it applies $\mathtt{RNM}$ to carefully constructed auxiliary scores that reduce sensitivity while appropriately penalizing high-sensitivity candidates. Its formal definition is given in Appendix A.

**Theorem 3.5** (Raskhodnikova & Smith, 2016). $\mathtt{GRNM}_\varepsilon(\mathcal{C}, q, \beta; D)$ *is $\varepsilon$-DP. Moreover, With probability $1 - \beta$, the output $u^*$ satisfies $q(u^*; D) \geq \max_{u \in \mathcal{C}}\{q(u; D) - (2\Delta_{q(u; \cdot)}/\varepsilon) \log(|\mathcal{C}|/\beta)\}$.*

We further utilize the generalized selection framework of Cohen et al. (2023) to select the best output from a collection of private subroutines. This framework provides tighter privacy guarantees than the composition theorems, as the privacy loss is independent of the total number of subroutines. See Appendix A for details.

**Submodular Functions.** Let $\mathcal{N}$ be a ground set of size $n$. For $f : 2^{\mathcal{N}} \to \mathbb{R}$, the marginal contribution of $u$ to $S$ is $f(u \mid S) \triangleq f(S \cup \{u\}) - f(S)$. $f$ is *monotone* if $f(S) \leq f(T)$ for all $S \subseteq T$, *non-negative* if $f(S) \geq 0$ for all $S$, and *submodular* if $f(u \mid S) \geq f(u \mid T)$ for all $S \subseteq T$ and $u \notin T$. In the DP setting, $f_D$ depends on sensitive dataset $D$, and its maximal sensitivity is given by

$$\Delta_f = \max_{S \subseteq \mathcal{N}, D \sim D'} |f_D(S) - f_{D'}(S)|.$$

For brevity, we omit the dataset subscript $D$ where it is clear from context.

**Submodular Maximization Subject to a Knapsack Constraint (SMK).** Given a submodular $f : 2^{\mathcal{N}} \to \mathbb{R}$, a positive cost function $c : \mathcal{N} \to \mathbb{R}_{>0}$, and budget $B > 0$, SMK aims to maximize $f(S)$ subject to the feasibility constraint $c(S) \triangleq \sum_{v \in S} c(v) \leq B$. We let $k$ denote the maximal cardinality of a feasible set and $c_{\min} \triangleq \min_{v \in \mathcal{N}} c(v)$. The *density* of $u \in \mathcal{N}$ w.r.t. $S \subseteq \mathcal{N} \setminus \{u\}$ is given by $f(u \mid S)/c(u)$. Following standard practice, we assume $c(v) \leq B$ for all $v \in \mathcal{N}$ and $B = 1$, as elements exceeding the budget are

---

[3]The PDF of $\mathrm{Exp}(\lambda)$ is $f(x; \lambda) = \lambda e^{-\lambda x}$ for $x \geq 0$.

never feasible and our algorithms are invariant under scaling of the costs. We assume access to $f$ via a *value oracle* that returns $f(S)$ for a given $S$. The total number of oracle queries performed by an algorithm serves as a proxy for its time complexity.

# 4. Challenges and Results

In this section, we outline the technical challenges of DP SMK, describe our proposed resolutions, and state our main results.

## 4.1. Monotone Objective Functions

To obtain a combinatorial algorithm with improved utility, we build on the $\mathtt{Two\text{-}Guess\text{-}Greedy}$ algorithm of (Feldman et al., 2023). Yet, its privatization is more involved than that of the standard greedy algorithm for cardinality constraints.

**Scores with Varying Sensitivity.** In the cardinality-constrained setting, differential privacy is typically achieved by replacing the greedy selection step with the Exponential Mechanism (McSherry & Talwar, 2007), a technique that yields strong guarantees in that context (Gupta et al., 2010; Mitrovic et al., 2017). However, this direct substitution is suboptimal for knapsack constraints, as density scores exhibit varying sensitivities that depend on element costs. A naive application of standard selection mechanisms would incur an additive error proportional to $\Delta_f \cdot B/c_{\min}$, a uniform sensitivity bound for all density scores. Since $B/c_{\min} \geq k$, this analysis would introduce a higher-order dependence on $k$, the maximal size of a feasible solution.

To ensure the additive error is independent of the worst-case sensitivity of a density score, we leverage the selection mechanism of Raskhodnikova & Smith (2016), which provides fine-grained utility bounds with respect to individual candidate sensitivities. Using this guarantee, we show that our algorithm attains a smaller additive error proportional to $\Delta_f$. This dependence matches the cardinality-constrained setting and avoids looser worst-case bounds, despite the higher sensitivity of density scores.

**Composition Bottleneck.** Another challenge stems from the $O(n^2)$ iterations required by $\mathtt{Two\text{-}Guess\text{-}Greedy}$, which pose a formidable composition bottleneck. A standard analysis based on advanced composition (Dwork et al., 2014) inevitably yields an error that scales polynomially with $n$. Hence, bridging the gap to the polylogarithmic dependence achieved in state-of-the-art results for matroid and cardinality constraints requires a different approach.

We observe that the highly parallelizable structure of $\mathtt{Two\text{-}Guess\text{-}Greedy}$ is advantageous not only for efficiency, but also for privacy. Specifically, the algorithm aligns naturally with a selection framework where the best

output is chosen from a collection of DP subroutines. In our case, these subroutines are privatized adaptations of the density-greedy algorithm. Based on the generalized selection framework of Cohen et al. (2023), we randomize the number of times each enumeration step is invoked. This allows us to bypass the standard composition penalty and obtain an additive error that is polylogarithmic in $n$, a significant improvement over prior work.

**Main Results.** Our approach is formalized in Algorithm 2, for which we establish the guarantee stated in Theorem 4.1. We provide the algorithm and an overview of the analysis in Section 5, with the full proofs deferred to Appendix B.

**Theorem 4.1.** *Let $f_D : 2^{\mathcal{N}} \to \mathbb{R}_+$ be a monotone submodular function with sensitivity $\Delta$. Then, for any $\varepsilon, \beta \in (0, 1)$ and $\delta > 0$, there exists an $(\varepsilon, \delta)$-DP algorithm for SMK that returns a feasible set $S \subseteq \mathcal{N}$ such that, with probability at least $1 - \beta$,*

$$f(S) \geq (1 - \tfrac{1}{e}) \cdot f(\text{OPT}) - O\left(\tfrac{\Delta k^{1.5}}{\varepsilon} \sqrt{\log \tfrac{n}{\delta\beta}} \log \tfrac{n}{\beta}\right).$$

*Moreover, the algorithm makes $O(\beta^{-1} n^3 k)$ oracle queries.*

As we show in Appendix B, the query complexity in Theorem 4.1 can be further improved to $O(n^3 k \log \beta^{-1})$ at the cost of an additional $\log(\beta^{-1})$ factor in the error term.

While `Two-Guess-Greedy` is, to our knowledge, the most practical existing approach for achieving a tight $(1-1/e)$ approximation, its computational cost remains high for large-scale instances. Thus, we additionally introduce a more efficient algorithm based on a single execution of a density-greedy variant (Yaroslavtsev et al., 2020). This result is stated in Theorem 4.2, with details provided in Appendix B.

**Theorem 4.2.** *Let $f_D : 2^{\mathcal{N}} \to \mathbb{R}_+$ be a monotone submodular function with sensitivity $\Delta$. Then, for any $\beta \in (0, 1)$ and $\varepsilon, \delta > 0$, there exists an $(\varepsilon, \delta)$-DP algorithm for SMK that returns a feasible set $S \subseteq \mathcal{N}$ such that, with probability at least $1 - \beta$,*

$$f(S) \geq \tfrac{1}{2} f(\text{OPT}) - O\left(\tfrac{\Delta k^{1.5}}{\varepsilon} \sqrt{\log \tfrac{1}{\delta}} \log \tfrac{n}{\beta}\right)$$

*Moreover, the algorithm makes $O(nk)$ oracle queries.*

### 4.2. Non-Monotone Objective Functions

At present, the most practical non-private approaches for non-monotone SMK combine density-greedy selection with randomized sampling by which selected elements are discarded with some fixed probability (Han et al., 2021; Amanatidis et al., 2020), ensuring high utility in expectation. Our proposed algorithm follows this paradigm, but its privatization introduces three primary obstacles.

**Composition Bottleneck.** The uncertainty in the number of selection steps creates a challenge for maintaining a tight

privacy analysis. Since selected elements can be discarded, the algorithm may perform up to $n$ selections. A standard application of advanced composition would yield an error term scaling with $\sqrt{n}$, rather than the desired logarithmic dependence. Moreover, unlike the non-private setting, initial subsampling of the ground set is not equivalent to post-selection discarding, since the output distribution of a DP selection mechanism depends on its entire candidate set.

Instead, we derive a concentration bound for the number of private selections, showing that the number of required private selections is stochastically dominated by a sum of independent geometric random variables that concentrates around $O(k)$. This ensures that, with high probability, the algorithm does not exceed the composition privacy budget.

**Robustness to Negative Marginal Gains.** Beyond composition, the non-monotone setting presents a challenge for private greedy selection. Analyses for non-monotone SMK, such as that of Han et al. (2021), rely on selecting elements with strictly positive marginal gain to ensure progress. In the DP setting, however, additive noise can cause elements with negative marginal contributions to be selected.

Instead of halting when no unselected elements with positive marginal gain exist, we continue adding feasible elements. Then, selecting the approximately best observed solution using `RNM` allows us to focus the analysis on early iterations, where feasible elements with sufficiently large gains exist. By extending the analysis of Han et al. (2021) to accommodate a *bounded additive slack*, and showing that elements with sufficiently large contribution are selected with high probability in early iterations, we obtain a $1/4$-approximation even when noise masks the sign of an element's marginal contribution.

**Scores with Varying Sensitivity.** Finally, the challenge of varying sensitivities identified in the monotone case (Section 4.1) persists here, as the density scores remain dependent on element costs. We resolve this in the same manner, leveraging `GRNM` to ensure that the additive error does not depend on the worst-case sensitivity of a density score.

**Main Result.** Our approach is formalized in Algorithm 3, for which we establish the guarantee in Theorem 4.3. We present the algorithm and analysis in Section 6, with full proofs deferred to Appendix C.

**Theorem 4.3.** *Let $f_D : 2^{\mathcal{N}} \to \mathbb{R}_+$ be a submodular function with sensitivity $\Delta$. Then, for any $\varepsilon, \beta \in (0, 1)$ and $\delta > 0$, there exists an $(\varepsilon, \delta)$-DP algorithm for SMK that returns a feasible set $S \subseteq \mathcal{N}$ such that, with probability at least $1 - \beta$,*

$$\mathbb{E}[f(S)] \geq \tfrac{1}{4} f(\text{OPT}) - O\left(\tfrac{\Delta k}{\varepsilon} \sqrt{(k + \log \tfrac{1}{\delta}) \log \tfrac{1}{\delta}} \log \tfrac{n}{\beta}\right).$$

*Moreover, the algorithm makes $O(nk)$ oracle queries.*

# 5. Monotone Objective Functions

In this section, we establish the guarantees stated in Theorem 4.1. The result follows by combining the complexity, privacy, and utility guarantees from Lemmas 5.1, 5.2 and 5.6. All omitted proofs are provided in Appendix B.

We introduce two components. The first, formulated in Algorithm 1 is a DP adaptation of the density-greedy algorithm that utilizes GRNM for greedy element selections. This serves as a core subroutine for the second component, Algorithm 2, which implements a privatized Two-Guess-Greedy algorithm. Specifically, Algorithm 2 samples $p \in [0, 1]$ uniformly and iterates over every subset $Y \subseteq \mathcal{N}$ of size $|Y| \leq 2$. For each $Y$, it performs $3/\beta$ iterations, each skipped independently with probability $p$. In non-skipped iterations where $|Y| = 2$, Algorithm 1 is executed on the residual instance defined by the ground set $\mathcal{N} \setminus Y$, remaining budget $B - c(Y)$, and marginal objective $h_D(S) = f_D(S \mid Y)$. The value of the resulting solution $S_Y \cup Y$ is then privatized using the Laplace mechanism. Finally, the candidate with the highest noisy value is returned.

---

**Algorithm 1** DP-Density-Greedy

**Input:** Dataset $D$; Submodular function $h_D : 2^{\mathcal{N}} \to \mathbb{R}_+$; Budget $B$; Parameters $\beta, \varepsilon \in (0, 1]$

1 Set $S_0 \leftarrow \varnothing$ and $i \leftarrow 0$.
2 **while** $\exists u \in \mathcal{N} \setminus S_i$ *such that* $c(S_i \cup \{u\}) \leq B$ **do**
3     Let $\mathcal{C}_i$ be the set of elements satisfying the condition stated in Line 2.
4     Define $q_i(u; D) = \frac{h_D(u|S_i)}{c(u)}$ for all $u \in \mathcal{C}_i$.
5     Compute $u_{i+1} \leftarrow \text{GRNM}_\varepsilon(\mathcal{C}_i, q_i, \beta/k; D)$
6     Let $S_{i+1} \leftarrow S_i \cup \{u_{i+1}\}$.
7     Update $i \leftarrow i + 1$.
8 **return** $S_i$

---

The query complexity of Algorithm 2 is stated next.

**Lemma 5.1.** *Algorithm 2 makes* $O(\beta^{-1}n^3k)$ *oracle queries.*

**Privacy Analysis.** The privacy guarantee of Algorithm 2 relies on a structural alignment with the generalized selection framework of (Cohen et al., 2023). This ensures a global privacy loss independent of the number of enumeration steps, complemented by a composition analysis for the individual invocations of Algorithm 1.

**Lemma 5.2.** *For any* $\varepsilon, \delta \in (0, 1]$*, Algorithm 2 is* $(\varepsilon, \delta)$*-DP.*

*Proof Sketch.* By advanced composition (Theorem 3.2), each invocation of Algorithm 1 is $(\varepsilon/6, \delta')$-DP, where $\delta' = O(\delta\beta/n^2)$. Consider a mechanism $\mathcal{M}_Y$ that executes Algorithm 1 on the residual instance defined by $Y$ and outputs $(S_Y, z_{Y,j})$ where $z_{Y,j} = f(S_Y \cup Y) + \text{Lap}(6\Delta_f/\varepsilon_0)$. By

basic composition, each $\mathcal{M}_Y$ is $(\varepsilon/3, \delta')$-DP. Finally, Algorithm 2 can be viewed as an instantiation of the Selection framework from (Cohen et al., 2023) applied to the collection $\{\mathcal{M}_Y \mid Y \subseteq \mathcal{N}, |Y| \leq 2\}$. Under our choice of parameters, the entire algorithm is $(\varepsilon, \delta)$-DP. $\square$

---

**Algorithm 2** DP-2GG (DP-Two-Guess-Greedy)

**Input:** Dataset $D$; Submodular function $f_D : 2^{\mathcal{N}} \to \mathbb{R}_+$; Budget $B$; Parameters $\beta, \varepsilon, \delta \in (0, 1]$

1 Sample $p \in [0, 1]$ uniformly; set $\Omega \leftarrow \varnothing$.
2 Let $\varepsilon_0 \leftarrow O(\varepsilon/\sqrt{k \log \frac{n}{\delta\beta}})$ // `Exact expression in the proof of Lemma B.2`
3 **foreach** $Y \subseteq \mathcal{N}$ *where* $|Y| \leq 2, c(Y) \leq B$ **do**
4     **foreach** $j = 1, \ldots, \lceil 3/\beta \rceil$ **do**
5         **with probability** $p$ **do**
6             **if** $|Y| = 2$ **then** let $S_{Y,j}$ be the output of Algorithm 1 on the residual instance defined by $Y$ with parameters $\beta/3, \varepsilon_0/6$.
7             **else** $S_{Y,j} \leftarrow \varnothing$.
8             Let $S'_{Y,j} \leftarrow S_{Y,j} \cup Y$
9             Let $z_{Y,j} \leftarrow f_D(S'_{Y,j}) + \text{Lap}\left(\frac{6\Delta_f}{\varepsilon}\right)$
10             Update $\Omega \leftarrow \Omega \cup \{(Y, j)\}$
11 **if** $\Omega \neq \varnothing$ **then** let $(Y^*, j^*) \leftarrow \text{argmax}_{(Y,j) \in \Omega} z_{Y,j}$ and $S^* \leftarrow S'_{Y^*,j^*}$
12 **else** $S^* \leftarrow \emptyset$
13 **return** $S^*$

---

**Utility Analysis.** The proof extends the analysis of Feldman et al. (2023) to address the error introduced by our DP adaptation. Assume that $|\text{OPT}| > 2$. The case $|\text{OPT}| \leq 2$ is addressed separately in Appendix B, as in this case the algorithm considers an optimal solution as a candidate in Line 11. While density-greedy does not provide a constant-factor approximation for SMK directly, a tight $(1 - 1/e)$-approximation is achieved when applied to a residual instance defined by $Y^*$, a maximum-value subset of OPT of size two (Feldman et al., 2023). Accordingly, we first analyze the execution of Algorithm 1 on this instance, beginning with a bound on the sensitivity of density scores.

**Lemma 5.3.** *For every iteration $i$ of Algorithm 1 and $u \in \mathcal{C}_i$, the quality score $q_i(u; D)$ has sensitivity $2\Delta_f/c(u)$.*

Let $o_m$ be an element of maximum cost in $\text{OPT} \setminus Y^*$, which is feasible in the residual instance and is non-empty by assumption. Let $S_1, \ldots, S_\ell$ denote the sequence of partial solutions produced by Algorithm 1. Define $T$ as the smallest index in $\{1, \ldots, \ell - 1\}$ for which the cost of $S_i$ exceeds $1 - c(Y^*) - c(o_m)$, or $T = \ell$ if no such index exists. The next lemma characterizes the progress made prior to iteration $T$ and leverages GRNM to obtain an error term that does not depend on the sensitivity of the density scores.

**Lemma 5.4.** *Let $Y^* \subseteq \mathrm{OPT}$ be a maximum-value subset of size $2$. Suppose Algorithm 1 is instantiated with parameters $\varepsilon$ and $\beta$ and executed on the residual instance defined by $Y^*$. Then, with probability $1 - \beta$, for every $i = 0, \ldots, T - 1$:*

$$(1 - c(Y^*) - c(o_m)) \cdot \frac{h(u_{i+1} \mid S_i)}{c(u_{i+1})} \geq$$

$$h(\mathrm{OPT} \setminus Y^*) - h(S_i) - h(o_m) - O\Big(\frac{k\Delta_f}{\varepsilon} \log \frac{n}{\beta}\Big).$$

We use Lemma 5.4 to show that when $S_i$ is far from optimal, each greedy step decreases the residual gap by a multiplicative factor with high probability. Unrolling this recurrence over the first $T$ iterations yields exponential decay of the residual term as a function of the accumulated cost $c(S_T)$. By the definition of $T$, the accumulated cost is sufficiently high to imply a $(1 - 1/e)$-approximation.

**Lemma 5.5.** *Suppose Algorithm 1 is executed on the residual instance defined by $Y^*$. Then, with probability $1 - \beta$, the returned set $S_\ell$ satisfies: $f(Y^* \cup S_\ell) \geq (1 - \frac{1}{e}) \cdot f(\mathrm{OPT}) - O\Big(\frac{k\Delta_f}{\varepsilon} \log \frac{n}{\beta}\Big)$.*

We now derive the utility bound. Notably, the guarantee holds even when $|\mathrm{OPT}| \leq 2$.

**Lemma 5.6.** *For any $\varepsilon, \delta > 0$, $\beta \in (0, 1)$, Algorithm 2 outputs a feasible $S^*$ such that, with probability $1 - \beta$,*

$$f(S^*) \geq (1 - \tfrac{1}{e}) \cdot f(\mathrm{OPT}) - O\Big(\frac{\Delta k^{1.5}}{\varepsilon} \sqrt{\log \frac{n}{\delta\beta}} \log \frac{n}{\beta}\Big).$$

*Proof sketch.* Consider an event $\mathcal{E}$ where: *(i)* Algorithm 1 is invoked for the guess $Y^*$, *(ii)* the event in Lemma 5.5 occurs, and (iii) all Laplace noises are bounded by $O(\frac{\Delta_f}{\varepsilon_0} \log \frac{n}{\beta})$. We show $\mathbf{Pr}[\mathcal{E}] \geq 1 - \beta$. Conditioned on $\mathcal{E}$, at least one candidate achieves a $(1 - 1/e)$-approximation up to a bounded additive error $O(\frac{k\Delta_f}{\varepsilon_0} \log \frac{n}{\beta})$, and the final selection step (Line 11) preserves this guarantee. Substituting $\varepsilon_0$ as set in Line 2 yields the bound. $\square$

So far, we have presented the key ideas behind the proof of Theorem 4.1. We conclude this section with a brief overview of our approach for Theorem 4.2, deferring the full algorithmic details and analysis to Section B.2.

**A Faster Algorithm.** We propose DP-DG$^+$ (Algorithm 7), a variant of Algorithm 1 that is executed once with the objective $f_D$, and uses RNM to select the best solution among: *(i)* the output of Algorithm 1, *(ii)* all feasible single-element extensions of partial solutions observed throughout the execution of Algorithm 1, and *(iii)* all singletons. The utility analysis follows an argument analogous to Lemma 5.4, but considers the element $o_m \in \mathrm{OPT}$ of maximal cost instead of $o_m \in \mathrm{OPT} \setminus Y^*$. This yields a progress inequality

for early iterations in which the total cost does not exceed $1 - c(o_m)$. Specifically, we show:

$$\frac{c(u_{i+1})}{1 - c(o_m)} \cdot \left(\frac{f(\mathrm{OPT})}{2} - kE\right) \leq f(S_{i+1}) - f(S_i),$$

where $E = O\Big(\frac{\Delta_f}{\varepsilon_0} \log \frac{n}{\beta}\Big)$. Summing this across all iterations leads to the bound stated in Theorem 4.2.

## 6. Non-Monotone Objective Functions

In this section, we present our algorithm for the non-monotone setting and outline the proof for Theorem 4.3. The result follows by combining the complexity, privacy, and utility guarantees established in Lemmas 6.1, 6.3 and 6.5. All omitted proofs are provided in Appendix C.

Algorithm 3 is based on the non-private SmkRan algorithm by Han et al. (2021). SmkRan maintains two solution sequences, $S_j$ and $S_j^*$. In each round $j$, it selects an element $u_j^*$ with maximum marginal gain and an element $u_{j+1}$ with maximum density. $S_j^*$ is set to $S_j \cup \{u_j^*\}$ if $u_j^*$ has positive marginal gain; otherwise, $S_j^* = S_j$. Similarly, $u_{j+1}$ is added to $S_j$ with probability $1/2$ only if its marginal gain is positive. The algorithm returns the best candidate among the final $S_j$ and all encountered $S_j^*$.

Our adaptation departs from this approach in several critical aspects. First, we perform only *density*-based selections using GRNM. The selected element is then added to the current solution with probability $1/2$; otherwise, it is discarded. Second, instead of testing for positive marginal gains, we use RNM to select the best candidate among all encountered partial solutions and their feasible single-element extensions. Critically, we extend the analysis of Han et al. (2021) to accommodate a controlled slack for slightly negative values, and prove that elements selected in *early* iterations provide sufficiently high marginal contributions with high probability. Finally, we derive a concentration bound on the total number of iterations to overcome the composition bottleneck described in Section 4.2.

The query complexity of Algorithm 3 is given by the following lemma.

**Lemma 6.1.** *Algorithm 3 makes $O(nk)$ oracle queries.*

**Privacy Analysis.** The privacy guarantee relies on a high-probability bound on the total number of iterations. We observe that the number of iterations between two consecutive element inclusions is stochastically dominated by a geometric random variable with parameter $1/2$. By extending this coupling to the entire execution, we establish that the total iteration count is dominated by a sum of independent geometric random variables, which concentrates strongly around $O(k)$.

---

**Algorithm 3** DP-SDG (DP-Subsample-Density-Greedy)

---

**Input:** Dataset $D$; Submodular function $f_D : 2^{\mathcal{N}} \to \mathbb{R}_+$; Budget $B$; Parameters $\beta, \varepsilon, \delta \in (0, 1]$.

1  Let $\varepsilon_0 \leftarrow O\left(\varepsilon / \sqrt{(k + \log(1/\delta)) \log(1/\delta)}\right)$    // Exact expression in the proof of Lemma C.6

2  Let $S_0 \leftarrow \varnothing$ and $i \leftarrow 0$.

3  **while** $\exists v \in \mathcal{N} \setminus \{u_1, \ldots, u_i\}$ *such that* $c(S_i + v) \leq B$ **do**

4      Let $\mathcal{C}_i$ be the set of elements satisfying the condition stated in Line 3.

5      Define $q_i(u; D) = \frac{f_D(v|S_i)}{c(v)}$ for all $v \in \mathcal{C}_i$.

6      Compute $u_{i+1} \leftarrow \mathtt{GRNM}_{\varepsilon_0}(\mathcal{C}_i, q_i, \beta/(2n); D)$

7      **with probability** $1/2$ **do**

8         $\lfloor \; S_{i+1} \leftarrow S_i \cup \{u_{i+1}\}$ .

9      **else** $S_{i+1} \leftarrow S_i$

10     Update $i \leftarrow i + 1$.

11  Let $\mathcal{S} \leftarrow \{S_{i'} \mid 0 \leq i' \leq i\} \cup \{S_{i'} \cup \{v\} \mid 0 \leq i' \leq i, \; v \in \mathcal{N}, \; c(S_{i'} + v) \leq B\}$

12  Let $S^* \leftarrow \mathtt{RNM}_{\varepsilon/2}(\mathcal{S}, f; D)$

13  **return** $S^*$

---

**Lemma 6.2.** *With probability at least* $1 - \delta$, *Algorithm 3 executes the while loop (Lines 3–10) at most* $O(k + \log(1/\delta))$ *times.*

Intuitively, the probability that the number of iterations, and hence the number of required composition steps, exceed $O(k + \log(1/\delta))$ can be absorbed into the privacy failure probability $\delta$. Combining this with the composition theorem (Theorem 3.2) yields our privacy result.

**Lemma 6.3.** *For any* $\varepsilon, \delta \in (0, 1]$, *Algorithm 3 is* $(\varepsilon, \delta)$-*DP.*

**Utility Analysis.** Our approach extends the analysis of Han et al. (2021). We first define a high-probability event $\mathcal{E}$ under which the DP-induced error remains bounded. The formal definition of $\mathcal{E}$ is deferred to the appendix, where we prove that $\mathbf{Pr}[\mathcal{E}] \geq 1 - \beta$. In particular, the occurrence of $\mathcal{E}$ guarantees that:

(i) For every $v \in \mathcal{C}_i$, the element $u_{i+1}$ obtained in Line 6 satisfies $\frac{f_D(u_{i+1}|S_i)}{c(u_{i+1})} \geq \frac{f_D(v|S_i)}{c(v)} - O\left(\frac{\Delta_f}{c(v)\varepsilon_0} \log \frac{n}{\beta}\right)$,

(ii) the set $S^*$ selected in Line 12 satisfies $f(S^*) \geq \max_{S \in \mathcal{S}} f(S) - O(\frac{\Delta_f}{\varepsilon} \log \frac{n}{\beta})$.

We analyze the expected utility conditioned on the occurrence of $\mathcal{E}$, starting by introducing notation. Let $o_m$ be an element of OPT with maximal cost. Let $S_0, \ldots, S_r$ be the partial solutions produced by Algorithm 3, where $r$ is the earliest iteration such that either the algorithm terminates, or there is no element $u \in \mathrm{OPT} \setminus S_r$ such that $S_r \cup \{u\}$ is feasible and $f(u \mid S_r) = \Omega(\frac{\Delta_f}{\varepsilon} \log \frac{n}{\beta})$.

Additionally, let $T$ be the smallest index $i < r$ such that $c(S_i \cup \{u_{i+1}\}) > c(\mathrm{OPT} \setminus \{o_m\})$, and set $T = r$ if no such index exists. We consider two disjoint subsets of OPT: $O_{\leq T}$ contains elements selected as $u_i$ for $i \leq T$, and $O_{>T}$ contains elements not in $O_{\leq T}$ with marginal contribution to $S_T$ at least $\Omega(\frac{\Delta_f}{\varepsilon_0} \log \frac{n}{\beta})$.

The following lemma bounds the total marginal contribution from elements in $\mathrm{OPT} \setminus \{o_m\}$ in terms of marginal gains from selected elements, discarded OPT elements, and a DP-induced error term. Crucially, it leverages Theorem 3.5 to ensure this error remains independent of the sensitivities of the density scores. The analysis relies on a delicate fractional mapping from Han et al. (2021), which allocates the cost of $O_{>T} \setminus \{o_m\}$ elements onto elements of the algorithm's partial solution. This mapping allows the proof to upper-bound the total marginal gain of those optimal elements in terms of selected elements.

**Lemma 6.4.** *For each* $u \in V$, *let* $X_u$ *be an indicator for the event that* $u \in O_{\leq T} \setminus (S_T \cup \{o_m\})$ *or* $u \in S_T \setminus (\mathrm{OPT} \setminus \{o_m\})$. *Then,*

$$f(S_T \cup \mathrm{OPT}) \leq f(S_T \cup \{o_m\}) + \sum_{u \in \mathcal{N}} X_u f(u \mid S_u)$$
$$+ f(S^*) - f(S_T) + O\left(\frac{k\Delta_f}{\varepsilon_0} \log \frac{n}{\beta}\right).$$

The utility guarantee of Algorithm 3 is stated next.

**Lemma 6.5.** *For any* $\varepsilon, \delta > 0$, $\beta \in (0, 1)$, *Algorithm 3 outputs a feasible set* $S^*$ *such that, with probability* $1 - \beta$,

$$\mathbb{E}[f(S^*)] \geq \tfrac{1}{4} f(\mathrm{OPT}) - O\left(\frac{\Delta k}{\varepsilon} \sqrt{(k + \log \frac{1}{\delta}) \log \frac{1}{\delta}} \log \frac{n}{\beta}\right).$$

*Proof Sketch.* We leverage a lemma from (Han et al., 2021) to show that $\mathbb{E}[f(S_T)] = \mathbb{E}[\sum_{u \in \mathcal{N}} X_u \cdot f(u \mid S_u)]$. Combining this equality with Lemma 6.4 yields the upper-bound

$$\mathbb{E}[f(S_T \cup \mathrm{OPT})] \leq 2\mathbb{E}[f(S^*)] + O\left(\frac{k\Delta_f}{\varepsilon_0} \log \frac{n}{\beta}\right).$$

On the other hand, a key lemma of Buchbinder et al. (2014) for non-monotone submodular functions implies the lower-bound $\mathbb{E}[f(S_T \cup \mathrm{OPT})] \geq \frac{1}{2} f(\mathrm{OPT})$, as each element is included in $S_T$ with probability at most $1/2$. Rearranging these inequalities and substituting $\varepsilon_0$ as set in Line 1 completes the proof. $\square$

## 7. Empirical Demonstration

We demonstrate the effectiveness of our proposed algorithms on a ride-sharing optimization task, a standard benchmark from prior work (Mitrovic et al., 2017; Chaturvedi et al., 2021). We examine the effect of $\varepsilon$ on utility (Figure 1) and the effect of $n$ on the number of oracle queries

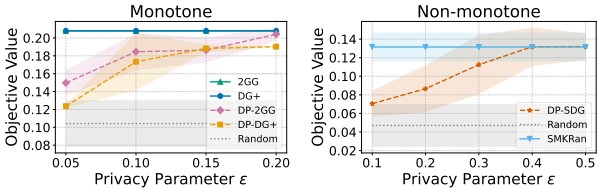

*Figure 1.* Monotone (left) and non-monotone (right) objective value for varying $\varepsilon$.

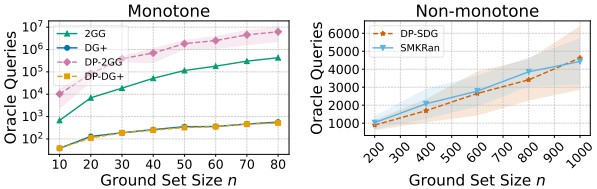

*Figure 2.* Monotone (left, log-scale) and non-monotone (right) number of queries for varying $n$.

(Figure 2). Appendix D includes results on the impact of the budget $B$ and higher-scale results for DP-DG$^+$. Our implementation is available online.[4]

**Dataset.** Following Mitrovic et al. (2017; 2018), we use a dataset $D$ of $m = 100,000$ Uber pickup locations in Manhattan (FiveThirtyEight, 2014), where each record is a (latitude, longitude) coordinate pair. The goal is to select a subset of waiting locations for idle drivers from a grid $\mathcal{N}$ of $n$ candidate points while ensuring differential privacy for individuals in the dataset.[5] We further assign each candidate $b \in \mathcal{N}$ a cost $c(b) = 1 + \frac{19}{\max(d_{\mathrm{km}}(b, p_{\mathrm{PoI}}), 1)} \in [1, 20]$, where $d_{\mathrm{km}}$ denotes distance in kilometers and $p_{\mathrm{PoI}}$ is a fixed point in Midtown Manhattan.[6] This induces a location premium that decays with distance from the city center.

**Objective Functions.** For a pickup point $a = (x_a, y_a)$ and a grid point $b = (x_b, y_b)$, we adopt the convenience score $s(a, b) = 2 - \frac{2}{1 + e^{-200\|a-b\|_1}}$ from Mitrovic et al. (2018). In the monotone setting, we maximize $f_D(S) = \frac{1}{m} \sum_{a \in D} \max_{b \in S} s(a, b)$, a monotone submodular function with sensitivity $1/m$. While $f_D$ measures coverage, it does not reward spatial diversity and may yield solutions concentrated in a small area. Thus, in the non-monotone setting, we maximize $g_D(S) = f_D(S) + \frac{\lambda}{nk} \sum_{b' \in \mathcal{N} \setminus S} \sum_{b \in S} s(b', b)$, fixing $\lambda = 0.1$. Since the added diversity term is a graph-cut function independent of $D$, $g_D$ is a non-monotone submodular function that retains sensitivity $1/m$.

**Algorithms.** In the monotone setting, we evaluate DP-2GG (Algorithm 2) and DP-DG$^+$ (Algorithm 7) against their non-

private counterparts Two-Guess-Greedy (2GG) (Feldman et al., 2023) and Density-Greedy$^+$ (DG$^+$) (Yaroslavtsev et al., 2020), while in the non-monotone setting we compare DP-SDG (Algorithm 3) with the non-private SmkRan (Han et al., 2021). In both settings, we include Random, a trivial DP baseline that processes elements in a random order, adding each to the solution if it maintains feasibility.

**Experimental Settings.** We use by default $\varepsilon = 1$ and $\delta = m^{-1.5}$. DP-SDG and DP-DG$^+$ are evaluated with defaults $n = 100, B = 20$ ($k = 8$). To accommodate the more query-intensive DP-2GG, experiments including it use a default of $n = 30, B = 40$ ($k = 12$). While we have used advanced composition in our theoretical analyses to yield asymptotically lower noise, constant factors may favor basic composition in some regimes. We set the per-iteration privacy parameter $\varepsilon_0$ according to the composition rule (basic or advanced) that achieves $(\varepsilon, \delta)$-DP with the smallest noise scale. All results are averaged over 10 runs and shaded regions indicate one standard deviation.

**Results.** Figure 1 shows that DP-2GG and DP-DG$^+$ achieve utility comparable to their non-private counterparts, substantially outperforming Random for $\varepsilon \geq 0.1$. At $\varepsilon = 0.2$, DP-2GG is within 2% of 2GG and DP-DG$^+$ within 8%, whereas Random is 49% lower. Moreover, DP-SDG attains utility close to that of the non-private SmkRan; at $\varepsilon = 0.4$, it is within 0.1%, while Random is 61% lower. Figure 2 shows that DP-DG$^+$ is significantly more scalable than DP-2GG, which requires $6000\times$ more queries at $n = 60$. Furthermore, DP-SDG exhibits linear growth in the number of queries with $n$, underscoring its practical efficiency. Similar trends for DP-DG$^+$ are reported in Appendix D.

# 8. Conclusion

We presented differentially private algorithms for maximizing monotone and non-monotone submodular functions subject to a knapsack constraint. Our results improve both utility and query complexity in the monotone setting and establish the first provable guarantees for the non-monotone setting. There are several interesting directions for future research. A primary question is whether an error term with better dependence on $k$ is achievable with polynomial query complexity under the bounded sensitivity assumption, a problem that remains unresolved even for cardinality constraints. Furthermore, while a $(1/2 - \eta)$-approximation can be achieved using $O_\eta(n)$ queries in the non-private setting via decreasing thresholds (Li et al., 2022), adapting such techniques to the DP setting without worsening the dependence on $k$ appears non-trivial. Finally, extending our techniques to settings combining knapsack and matroid constraints (Badanidiyuru & Vondrák, 2014; Mirzasoleiman et al., 2016) is a natural direction for future work.

---

[4] https://github.com/ronzadi/dp-smk.

[5] Assuming each pickup corresponds to a distinct individual.

[6] Specifically, we use Penn Station as the reference location.

## Acknowledgments

This research was supported by the Israel Science Foundation (ISF) under grant 2707/22 of the Breakthrough Research Grant (BRG) Program.

## Impact Statement

This work advances the area of private optimization. There might be potential societal consequences of our work, none of which we feel must be specifically highlighted here.

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

# A. Generalized Private Selection

In this section, we provide the technical background for the generalized DP selection mechanisms utilized throughout this work. We first address the problem of private selection under varying sensitivities in Section A.1. Subsequently, in Section A.2, we discuss the selection framework used to select the best candidate from a collection of DP releases.

## A.1. Report Noisy Max for Scores with Varying Sensitivity

Consider a finite set $\mathcal{C}$, where each element $u \in \mathcal{C}$ is associated with a quality score $q(u; D) : \mathcal{X}^m \to \mathbb{R}$. A fundamental task in differential privacy is the private selection of an element $u \in \mathcal{C}$ that approximately maximizes the quality score $q(u; D)$ given a sensitive dataset $D \in \mathcal{X}^m$. One of the most widely used and computationally efficient mechanisms for this task is the *Report Noisy Max* mechanism with exponential noise (Dwork et al., 2014), denoted $\mathtt{RNM}_\varepsilon(\mathcal{C}, q; D)$. The mechanism perturbs each score with independent exponential noise and returns the candidate with the highest perturbed value.

*Remark* A.1. Report Noisy Max with exponential noise has been shown to be equivalent to the $\mathtt{Permute\text{-}and\text{-}Flip}$ mechanism (Ding et al., 2021), which dominates the Exponential Mechanism in terms of utility (McKenna & Sheldon, 2020). While we use exponential noise for this reason, our analysis is not strictly dependent on this choice; other distributions, such as Laplace noise (Dwork et al., 2014), would yield similar results.

---

**Algorithm 4** $\mathtt{RNM}_\varepsilon(\mathcal{C}, q; D)$ (Dwork et al., 2014)

---

**Input:** Dataset $D \in \mathcal{X}^m$; candidate set $\mathcal{C}$; for each $u \in \mathcal{C}$ a score function $q(u; D) : \mathcal{X}^m \to \mathbb{R}$ with sensitivity $\Delta_u$; privacy parameter $\varepsilon > 0$.

1 $\Delta \leftarrow \max_{u \in \mathcal{C}} \Delta_u$
2 **for** $u \in \mathcal{C}$ **do**
3 $\quad$ Sample $Z_u \leftarrow \mathrm{Exp}\left(\frac{\varepsilon}{2\Delta}\right)$ independently
4 **return** $\arg\max_{u \in \mathcal{C}}\{q(u; D) + Z_u\}$.

---

**Theorem A.2** (Restatement of Theorem 3.4, Dwork et al., 2014). *For any $\varepsilon > 0$, $\mathtt{RNM}_\varepsilon(\mathcal{C}, q; D)$ satisfies $\varepsilon$-differential privacy. Moreover, for any $\beta \in (0, 1)$, with probability at least $1 - \beta$, the output $u^*$ satisfies*

$$q(u^*; D) \geq \max_{u \in \mathcal{C}} q(u; D) - \frac{2\Delta}{\varepsilon} \log \frac{|\mathcal{C}|}{\beta}.$$

As the bound shows, the additive error of standard $\mathtt{RNM}$ scales with the maximum sensitivity across all candidates. In our setting, scores arise from density-based greedy selections, and this bound introduces a suboptimal dependence on element costs, as discussed in Section 4. This motivates the use of a more refined private selection mechanism that adapts to individual sensitivities.

We employ the generalized selection mechanism of (Raskhodnikova & Smith, 2016), which we denote by $\mathtt{GRNM}_\varepsilon(\mathcal{C}, q, \beta; D)$. The mechanism takes an additional failure probability parameter $\beta$ and internally invokes $\mathtt{RNM}$ on carefully constructed auxiliary scores. These scores are designed to reduce sensitivity while appropriately penalizing candidates with large individual sensitivities. Next, we provide the pseudocode and review the utility proof.

---

**Algorithm 5** $\mathtt{GRNM}_\varepsilon(\mathcal{C}, q, \beta; D)$ (Raskhodnikova & Smith, 2016)

---

**Input:** Dataset $D \in \mathcal{X}^m$; candidate set $\mathcal{C}$; for each $u \in \mathcal{C}$ a score function $q(u; D) : \mathcal{X}^m \to \mathbb{R}$ with sensitivity $\Delta_u$; parameters $\varepsilon > 0$ and $\beta \in (0, 1)$.

1 Set $t \leftarrow \frac{2}{\varepsilon} \log \frac{|\mathcal{C}|}{\beta}$
2 **for** $u \in \mathcal{C}$ **do**
3 $\quad$ Define $q'(u; D) \leftarrow \min_{v \in \mathcal{C}} \frac{(q(u;D) - t\Delta_u) - (q(v;D) - t\Delta_v)}{\Delta_u + \Delta_v}$
4 $\quad$ Sample $Z_u \leftarrow \mathrm{Exp}\left(\frac{\varepsilon}{2}\right)$ independently
5 **return** $\arg\max_{u \in \mathcal{C}}\{q'(u; D) + Z_u\}$.

---

**Theorem A.3** (Formal version of Theorem 3.5, Raskhodnikova & Smith, 2016). *Suppose that for each $u \in \mathcal{C}$, the score function $q(u; D) : \mathcal{X}^m \to \mathbb{R}$ has sensitivity $\Delta_u$ (with respect to its dataset argument). For any $\varepsilon > 0$ and $\beta \in (0, 1)$, the*

*mechanism* $\mathrm{GRNM}_\varepsilon(\mathcal{C}, q, \beta; D)$ *is $\varepsilon$-DP. Furthermore, with probability at least $1 - \beta$, the output $u^*$ satisfies*

$$q(u^*; D) \geq \max_{u \in \mathcal{C}} \left\{ q(u; D) - \frac{4\Delta_u}{\varepsilon} \log \frac{|\mathcal{C}|}{\beta} \right\}.$$

*Proof.* The privacy guarantee follows directly from the guarantee of RNM (Theorem A.2), since for each $u \in \mathcal{C}$, the score $q'(u; D)$ has sensitivity at most 1 (with respect to its dataset argument).

By a union bound and the tail of the exponential distribution, with probability at least $1 - \beta$ all noise variables are bounded by $t = \frac{2}{\varepsilon} \log \frac{|\mathcal{C}|}{\beta}$. Condition on this event. Let $u^*$ denote the selected element. Then for any $v \in \mathcal{C}$,

$$q'(u^*; D) \geq q'(v; D) - t,$$

since otherwise $q'(v; D) + Z_v > q'(u^*; D) + Z_{u^*}$, contradicting the choice of $u^*$.

Let $v' \in \mathcal{C}$ be the element maximizing $q(v'; D) - t\Delta_{v'}$. For all $u \in \mathcal{C}$, we have $q(v'; D) - t\Delta_{v'} \geq q(u; D) - t\Delta_u$, which implies

$$q'(v'; D) = \min_{u \in \mathcal{C}} \frac{(q(v'; D) - t\Delta_{v'}) - (q(u; D) - t\Delta_u)}{\Delta_{v'} + \Delta_u} \geq 0.$$

Combining the inequalities yields $q'(u^*; D) \geq -t$.

Finally, for any $u \in \mathcal{C}$, the definition of $q'$ gives

$$q'(u^*; D) \leq \frac{(q(u^*; D) - t\Delta_{u^*}) - (q(u; D) - t\Delta_u)}{\Delta_{u^*} + \Delta_u}.$$

Together with $q'(u^*; D) \geq -t$, this implies

$$q(u^*; D) \geq q(u; D) - 2t\Delta_u,$$

completing the proof. $\qquad\square$

*Remark* A.4. As the previous proof shows, instead of setting $t = \frac{2}{\varepsilon} \log \frac{|\mathcal{C}|}{\beta}$, one may use any upper bound $M \geq |\mathcal{C}|$, in which case, with probability at least $1 - \beta$, the output $u^*$ satisfies

$$q(u^*; D) \geq \max_{u \in \mathcal{C}} \left\{ q(u; D) - \frac{4\Delta_u}{\varepsilon} \log \frac{M}{\beta} \right\}.$$

In Appendix C, we take $t = \frac{2}{\varepsilon} \log \frac{n}{\beta}$, where $n$ denotes the size of the ground set, as this choice is used in our utility analysis in that section. Specifically, it allows us to sample all DP noises upfront, and focus our analysis on a randomized execution of Algorithm 3 with fixed realization for the DP mechanisms. Note that all candidate sets $\mathcal{C}_i$ arising in Algorithm 3 satisfy $|\mathcal{C}_i| \leq n$.

### A.2. Generalized Private Selection

A more general setting than the one considered in Section A.1 arises when we have a collection of $k$ mechanisms $\{\mathcal{M}_i\}_{i=1}^k$, each producing outputs in an ordered domain (e.g., scored solutions), and the goal is to select an approximately best solution produced by any $\mathcal{M}_i$, while incurring a privacy cost close to that of executing a single $\mathcal{M}_i$ and releasing its output. This problem is commonly referred to as *generalized private selection* (Liu & Talwar, 2019; Papernot & Steinke, 2022; Cohen et al., 2023), and it has found various applications in settings such as parameter tuning in machine learning. In this work, we adopt the framework of (Cohen et al., 2023) due to its simplicity and high utility when combined with our analysis.

The algorithm $\mathrm{Selection}_{\tau,\gamma}$ is parameterized by $\gamma \in (0, \infty)$ and $\tau \in \mathbb{N}$. Given $k$ private mechanisms $\{\mathcal{M}_i\}_{i=1}^k$ producing scored outputs, it first samples $p \in [0, 1]$ where $\mathbf{Pr}[p \leq x] = x^\gamma$ for all $x \in [0, 1]$. Then, for each $i \in [k]$, it runs $\mathcal{M}_i$ a random number of times drawn as $\mathrm{Bin}(\tau, p)$. Finally, it outputs the highest scored candidate observed across all invocations. The pseudocode is as follows:

---

**Algorithm 6** $\texttt{Selection}_{\tau,\gamma}$ (Cohen et al., 2023)

---

**Input:** Dataset $D$; Private mechanisms $\{\mathcal{M}_i\}_{i=1}^{k}$
1 Sample $p \in [0,1]$ where $\mathbf{Pr}[p \le x^\gamma] = x^\gamma$ for all $x \in [0,1]$
2 $S \leftarrow \varnothing$
3 **for** $i = 1, \ldots, k$ **do**
4      **for** $j = 1, \ldots, \tau$ **do**
5          **with probability** $p$ **do**
6              $s_{i,j} \leftarrow \mathcal{M}_i(D)$
7              $S \leftarrow S \cup \{s_{i,j}\}$

8 **return** $\operatorname{argmax}_{x \in S} x$

---

The privacy guarantee of this mechanism is stated next. Although Cohen et al., 2023 proves a more general result, the statement below is sufficient for our purposes.

**Theorem A.5** (Cohen et al., 2023). *Suppose each mechanism $\mathcal{M}_i$ is $(\varepsilon, \delta_i)$-DP with $\delta_i \ge 0$. Then, for any choice of $\tau \in \mathbb{N}$ and $\gamma \in (0, \infty)$, the algorithm $\texttt{Selection}_{\tau,\gamma}$ is $((2+\gamma)\varepsilon, \tau \sum_{i=1}^{k} \delta_i)$-DP.*

## B. Proofs from Section 5

In this section, we provide the missing proofs from Section 5 for monotone submodular maximization. We begin with the proof of Theorem 4.1. Then, we move on to present our faster algorithm in this setting and prove Theorem 4.2.

### B.1. Proof of Theorem 4.1

**Lemma B.1** (Lemma 5.1 restated). *Algorithm 2 makes $O(\beta^{-1} n^3 k)$ oracle queries.*

*Proof.* Each execution of Algorithm 1 consists of at most $k$ iterations of the while loop, as the cardinality of the solution increases by one and remains feasible after each iteration. Each iteration evaluates $O(n)$ marginal gains, resulting in $O(nk)$ oracle queries per execution. Algorithm 2 invokes Algorithm 1 at most $O(\beta^{-1})$ times for every subset of $\mathcal{N}$ with cardinality at most 2, totaling $O(n^2 \beta^{-1})$ invocations. Consequently, the total query complexity is $O(\beta^{-1} n^3 k)$. $\qquad \square$

### B.1.1. PRIVACY ANALYSIS

**Lemma B.2** (Lemma 5.2 restated). *For any $\varepsilon, \delta \in (0, 1]$, Algorithm 2 is $(\varepsilon, \delta)$-DP.*

*Proof.* Let $\delta' = \frac{\delta}{n^2 \lceil 3/\beta \rceil}$. We begin by analyzing the privacy guarantee of Algorithm 1, instantiated with privacy parameter $\varepsilon_0/6$, where

$$\varepsilon_0 = \frac{\varepsilon}{2\sqrt{2k \log(1/\delta')}}.$$

The algorithm invokes $\texttt{GRNM}$ sequentially at most $k$ times, each with parameter $\varepsilon_0/6$. By the privacy guarantee of $\texttt{GRNM}$ (Theorem A.3) and the advanced composition theorem (Theorem 3.2), it follows that Algorithm 1 is $(\varepsilon/6, \delta')$-DP.

We next consider any subset $Y \subseteq \mathcal{N}$ of size at most two and define a corresponding mechanism $\mathcal{M}_Y$. The mechanism first samples noise $z_Y \sim \text{Lap}(\frac{6\Delta_f}{\varepsilon})$. If $|Y| = 2$, it runs Algorithm 1 on the residual instance induced by $Y$, obtains its output $S_Y$, and outputs the pair $(S_Y, f(S_Y \cup Y) + z_Y)$. By the guarantee of the Laplace mechanism (Theorem 3.3), the privacy guarantee of Algorithm 1 established above, and basic composition (Theorem 3.2), $\mathcal{M}_Y$ is $(\varepsilon/3, \delta')$-DP. If $|Y| < 2$, the mechanism outputs $(Y, f(Y) + z_Y)$, which is $(\varepsilon/6, 0)$-DP and in particular $(\varepsilon/3, \delta')$-DP.

Finally, Algorithm 2 can be viewed as an instantiation of $\texttt{Selection}_{\tau,\gamma}$ applied to the collection of mechanisms

$$\{\mathcal{M}_Y \mid Y \subseteq \mathcal{N}, \ |Y| \le 2, \ c(Y) \le B\},$$

with parameters $\tau = \lceil 3/\beta \rceil$ and $\gamma = 1$. By the privacy guarantees of the generalized private selection procedure (Theorem A.5), the overall algorithm satisfies $(\varepsilon, \delta)$-DP, since

$$\tau \cdot \sum_Y \delta' \leq \tau n^2 \delta' = \delta.$$

$\square$

### B.1.2. Utility Analysis

We follow the proof outlined by Feldman et al., 2023. We begin by recalling the notation introduced in Section 5. Let OPT denote an optimal solution, and assume first that $|\text{OPT}| > 2$. Let $Y^* \subseteq \text{OPT}$ be a subset of size two with maximum value, and consider the residual instance defined by the ground set $\mathcal{N} \setminus Y^*$, budget $1 - c(Y^*)$, and objective function $h(S) = f(S \mid Y^*)$. Note that $h$ is a non-negative, monotone, and submodular function. Recall that we normalize the total budget to $B = 1$.

Let $O = \text{OPT} \setminus Y^*$, and let $o_m$ be an element in $O$ with maximum cost, i.e., $o_m \in \arg\max_{o \in O} c(o)$. Note that $O$ is feasible in the residual instance and is non-empty by the assumption that $|\text{OPT}| > 2$. Let $S_1, \ldots, S_\ell$ denote the sequence of partial solutions generated by Algorithm 1, where $S_\ell$ is the returned solution. Define:

$$T = \min\left\{i \mid (1 \leq i \leq \ell - 1 \text{ and } c(S_i) > 1 - c(Y^*) - c(o_m)) \text{ or } (i = \ell)\right\}.$$

Intuitively, $T$ is the first index among $i = 1, \ldots, \ell - 1$ such that the cost exceeds $1 - c(Y^*) - c(o_m)$, if such an index exists; otherwise, $T = \ell$.

**Lemma B.3** (Lemma 5.3 restated). *For every iteration $i$ of Algorithm 1 and $u \in \mathcal{C}_i$, the quality score $q_i(u; D)$ has sensitivity $2\Delta_f/c(u)$.*

*Proof.* Recall that for all $u \in \mathcal{C}_i$ we have $q_i(u; D) = \frac{h_D(u|S_i)}{c(u)}$, and $h_D(S) = f_D(S \mid Y)$ for some subset $Y \subseteq \mathcal{N}$. Expanding the definition of $h_D$, we obtain

$$q_i(u; D) = \frac{h_D(S_i \cup \{u\} \mid Y) - h_D(S_i \mid Y)}{c(u)} = \frac{f_D(S_i \cup Y \cup \{u\}) - f_D(S_i \cup Y)}{c(u)}.$$

Since $f_D(S)$ has sensitivity at most $\Delta_f$ for any $S \subseteq \mathcal{N}$, for any two neighboring datasets $D, D'$

$$\begin{aligned}
|q_i(u; D) - q_i(u; D')| &\leq \frac{1}{c(u)}\left|f_D(S_i \cup Y \cup \{u\}) - f_{D'}(S_i \cup Y \cup \{u\})\right| \\
&\quad + \frac{1}{c(u)}\left|f_{D'}(S_i \cup Y) - f_D(S_i \cup Y)\right| \\
&\leq \frac{2\Delta_f}{c(u)}.
\end{aligned}$$

$\square$

**Lemma B.4** (Formal version of Lemma 5.4). *Let $Y^* \subseteq \text{OPT}$ be a maximum-value subset of size 2. Suppose Algorithm 1 is instantiated with parameters $\varepsilon$ and $\beta$ and executed on the residual instance defined by $Y^*$, i.e., with ground set $\mathcal{N} \setminus Y^*$, budget $1 - c(Y^*)$, and objective $h(S) = f(S \mid Y^*)$. Then, with probability $1 - \beta$, for every $i = 0, \ldots, T - 1$:*

$$(1 - c(Y^*) - c(o_m)) \cdot \left(\frac{h(u_{i+1} \mid S_i)}{c(u_{i+1})}\right) \geq h(O) - h(S_i) - h(o_m) - \frac{8k\Delta_f}{\varepsilon} \log \frac{nk}{\beta}.$$

*Proof.* Since the sets $S_i$ remain feasible in all iterations, the number of iterations is at most $k$. By the utility guarantee of GRNM (Theorem A.3) and the sensitivity bound (Theorem B.3), in each iteration $i$, with probability at least $1 - \beta/k$, GRNM selects an element $u_{i+1}$ satisfying:

$$\frac{h(u_{i+1} \mid S_i)}{c(u_{i+1})} \geq \max_{u \in \mathcal{C}_i}\left\{\frac{h(u \mid S_i)}{c(u)} - \frac{8\Delta_f}{\varepsilon} \log \frac{nk}{\beta}\right\}. \tag{1}$$

By a union bound, this holds for all iterations with probability at least $1 - \beta$. We condition on this event for the remainder of the proof.

Let $R_i = S_i \cup \{o_m\}$ and let $E = \frac{8\Delta_f}{\varepsilon} \log \frac{nk}{\beta}$. We observe:

$$
\begin{aligned}
h(O) - h(S_i \cup \{o_m\}) &= h(O) - h(R_i) \\
&\leq h(O \cup R_i) - h(R_i) && \text{(Monotonicity)} \\
&\leq \sum_{s \in O \setminus R_i} h(s \mid R_i) && \text{(Submodularity)} \\
&= \sum_{s \in O \setminus R_i} c(s) \frac{h(s \mid R_i)}{c(s)} \\
&\leq \sum_{s \in O \setminus R_i} c(s) \cdot \left( \frac{h(u_{i+1} \mid S_i)}{c(u_{i+1})} + \frac{E}{c(s)} \right) && (2) \\
&= c(O \setminus R_i) \cdot \frac{h(u_{i+1} \mid S_i)}{c(u_{i+1})} + |O \setminus R_i| \cdot E \\
&\leq (c(O) - c(O \cap R_i)) \cdot \frac{h(u_{i+1} \mid S_i)}{c(u_{i+1})} + kE \\
&\leq (1 - c(Y^*) - c(o_m)) \cdot \frac{h(u_{i+1} \mid S_i)}{c(u_{i+1})} + kE. && (3)
\end{aligned}
$$

For inequality (2), note that since $i < T$, we have $c(S_i) + c(o_m) \leq 1 - c(Y^*)$. Thus, every $s \in O \setminus R_i$ can be added to $S_i$ without violating feasibility (i.e., $O \setminus R_i \subseteq C_i$), and the bound follows from (1) and the fact that $h(s \mid R_i) \leq h(s \mid S_i)$ by submodularity. The final inequality holds by $c(O) \leq 1 - c(Y^*)$ and $o_m \in O \cap R_i$.

By the submodularity and non-negativity of $h$ we have $h(S_i \cup \{o_m\}) \leq h(S_i) + h(o_m)$. Rearranging the terms in the established chain of inequalities then yields the lemma. $\square$

**Lemma B.5.** *Suppose Algorithm 1 is instantiated with parameters $\varepsilon$ and $\beta$, and executed on the residual instance defined as in Lemma B.4. Then, with probability $1 - \beta$, Then, with probability $1 - \beta$,*

$$
h(S_\ell) \geq (1 - \tfrac{1}{e}) \cdot \left( h(O) - h(o_m) \right) - \frac{8k\Delta_f}{\varepsilon} \log \frac{nk}{\beta}.
$$

*Proof.* Let us denote $E = \frac{8\Delta_f}{\varepsilon} \log \frac{nk}{\beta}$. We may assume without loss of generality that $h(S_i) < h(O) - h(o_m) - kE$ for every $0 \leq i \leq T$, since otherwise the lemma follows immediately from the monotonicity of $h$. Applying Lemma 5.4 for every $0 \leq i \leq T - 1$, we get

$$
\frac{h(S_{i+1}) - h(S_i)}{c(u_{i+1})} = \frac{h(u_{i+1} \mid S_i)}{c(u_{i+1})} \geq \frac{h(O) - h(S_i) - h(o_m) - kE}{1 - c(Y^*) - c(o_m)} \qquad \forall i \in \{0, \dots, T-1\}.
$$

Rearranging yields

$$
\begin{aligned}
h(O) - h(S_{i+1}) - h(o_m) - kE &\leq \left( 1 - \frac{c(u_{i+1})}{1 - c(Y^*) - c(o_m)} \right) [h(O) - h(S_i) - h(o_m) - kE] \\
&\leq e^{-c(u_{i+1})/(1 - c(Y^*) - c(o_m))} \cdot [h(O) - h(S_i) - h(o_m) - kE]
\end{aligned}
$$

Unraveling the last inequality for all $0 \leq i \leq T - 1$ gives

$$
\begin{aligned}
h(O) - h(S_T) - h(o_m) - kE &\leq \prod_{i=0}^{T-1} e^{-c(u_{i+1})/(1 - c(Y^*) - c(o_m))} \cdot [h(O) - h(S_0) - h(o_m) - kE] \\
&= e^{-c(S_T)/(1 - c(Y^*) - c(o_m))} \cdot [h(O) - h(S_0) - h(o_m) - kE] \\
&\leq e^{-1} \cdot [h(O) - h(o_m) - kE]
\end{aligned}
$$

To see why the last inequality holds, consider two cases. If $T < \ell$, then by definition of $T$ we have $c(S_T) > 1 - c(Y^*) - c(o_m)$. Now consider the case $T = \ell$. If $O \subseteq S_\ell$, then the inequality in the lemma holds by monotonicity of $h$. Otherwise, there exists an element $o \in O \setminus S_\ell$. Since Algorithm 1 has terminated at iteration $\ell$, no more elements can be added while maintaining feasibility in the residual instance, and therefore $c(S_\ell) > 1 - c(Y^*) - c(o) \geq 1 - c(Y^*) - c(o_m)$.

By rearranging we get

$$h(S_T) \geq (1 - \tfrac{1}{e})[h(O) - h(o_m) - kE]$$

The lemma follows by monotonicity.

$\square$

**Lemma B.6** (Formal version of Lemma 5.5)**.** *Suppose Algorithm 1 is instantiated with parameters $\varepsilon > 0$ and $\beta \in (0,1)$, and executed on the residual instance defined as in Lemma B.4. Then, with probability $1 - \beta$,*

$$f(Y^* \cup S_\ell) \geq (1 - \tfrac{1}{e}) \cdot f(\mathrm{OPT}) - \frac{8k\Delta_f}{\varepsilon} \log \frac{nk}{\beta}.$$

*Proof.* With probability $1 - \beta$, the event stated in Lemma 5.4 holds. Let $S_1, \ldots, S_\ell$ be a sequence of sets computed by the Algorithm 1 conditioned on this event. We have

$$f(Y^* \cup S_\ell) = h(S_\ell) + f(Y^*) \geq (1 - \tfrac{1}{e}) \cdot \big[h(\mathrm{OPT} \setminus Y^*) - h(o_m)\big] + f(Y^*) - \frac{8k\Delta_f}{\varepsilon} \log \frac{nk}{\beta}$$

$$= (1 - \tfrac{1}{e}) \cdot f(\mathrm{OPT}) + \tfrac{1}{e} \cdot f(Y^*) - (1 - \tfrac{1}{e})f(o_m \mid Y^*) - \frac{8k\Delta_f}{\varepsilon} \log \frac{nk}{\beta},$$

where the first inequality holds by Lemma B.5 Hence it remains to show that $\tfrac{1}{e}f(Y^*) \geq (1 - \tfrac{1}{e})f(o_m \mid Y^*)$. The proof is identical to that in (Feldman et al., 2023), and follows from the definition of $Y^*$ and submodularity:

$$\tfrac{1}{e} \cdot f(Y^*) \geq \tfrac{1}{e} \cdot \big[f(\{u_1, o_m\}) + f(\{u_2, o_m\}) - f(Y^*)\big]$$

$$= \tfrac{1}{e} \cdot \big[f(o_m \mid \{u_1\}) + f(o_m \mid \{u_2\}) + f(u_1) + f(u_2) - f(Y^*)\big]$$

$$\geq \tfrac{2}{e} \cdot f(o_m \mid Y^*)$$

$$\geq (1 - \tfrac{1}{e}) \cdot f(o_m \mid Y^*).$$

$\square$

The next lemma concludes the utility proof for Algorithm 2. We remove the assumption that $|\mathrm{OPT}| > 2$ and establish the guarantee for the general case.

**Lemma B.7** (Lemma 5.6 restated)**.** *For any $\varepsilon, \delta > 0$, $\beta \in (0,1)$, Algorithm 2 outputs a feasible $S^*$ such that, with probability $1 - \beta$,*

$$f(S) \geq (1 - \tfrac{1}{e}) \cdot f(\mathrm{OPT}) - O\left(\frac{k^{1.5}\Delta_f}{\varepsilon} \sqrt{\log \frac{n}{\delta\beta}} \log \frac{n}{\beta}\right).$$

*Proof.* We first consider the case $|\mathrm{OPT}| \geq 2$ and focus on the iteration of Algorithm 2 in which the algorithm guesses $Y^*$. In this iteration, the number of executions of Algorithm 1 is distributed uniformly over $\{0, \ldots, \lceil 3/\beta \rceil\}$. Indeed, since $p$ is sampled uniformly from $[0, 1]$ and the number of executions is distributed as $\mathrm{Bin}(\lceil 3/\beta \rceil, p)$, each outcome occurs with equal probability.[7] Consequently, Algorithm 1 is executed at least once with probability at least $1 - \beta/3$. We condition on this event.

If $|\mathrm{OPT}| > 2$, then by Lemma 5.5, with probability at least $1 - \beta/3$, the first execution of Algorithm 1 in this iteration outputs a set $S_{Y^*, j}$ satisfying

$$f(Y^* \cup S_{Y^*, j}) \geq (1 - \tfrac{1}{e}) \cdot f(\mathrm{OPT}) - \frac{48k\Delta_f}{\varepsilon_0} \log \frac{3nk}{\beta}. \tag{4}$$

---

[7]For all $m \in \{0, 1, \ldots, \tau\}$, $\mathbb{E}_{p \sim U[0,1]} \binom{\tau}{m} p^m (1-p)^{\tau-m} = 1/(\tau + 1)$ (Cohen et al., 2023).

If instead $|\mathrm{OPT}| = 2$, then $Y^* = \mathrm{OPT}$, and the same inequality follows immediately from the monotonicity of $f$. We therefore condition on the event that (4) holds.

Next, standard Laplace tail bounds and a union bound imply that, with probability at least $1 - \beta/3$, every Laplace noise sampled in Line 9 during the execution of Algorithm 2 has magnitude at most $\frac{6\Delta_f}{\varepsilon} \log \frac{3n}{\beta}$. Conditioning on this event, the set $S^*$ selected in Line 11 satisfies

$$f(S^*) \geq f(Y^* \cup S_{Y^*, j}) - \frac{12\Delta_f}{\varepsilon} \log \frac{3n}{\beta}.$$

Combining the above events and applying a union bound, we conclude that with probability at least $1 - \beta$,

$$f(S^*) \geq (1 - \tfrac{1}{e}) \cdot f(\mathrm{OPT}) - \frac{12\Delta_f}{\varepsilon} \log \frac{3n}{\beta} - \frac{48k\Delta_f}{\varepsilon_0} \log \frac{3nk}{\beta}$$

$$\geq (1 - \tfrac{1}{e}) \cdot f(\mathrm{OPT}) - O\left(\frac{k\Delta_f}{\varepsilon_0} \log \frac{n}{\beta}\right),$$

where we have used that $k \leq n$. The inequality in the lemma follows by substitution the value for $\varepsilon_0$ defined in the proof of Lemma B.2. In particular, $\varepsilon_0 = O\left(\varepsilon / \sqrt{k \log \frac{n}{\delta\beta}}\right)$.

It remains to consider the case $|\mathrm{OPT}| = 1$. In this case, one iteration of the algorithm guesses $Y = \mathrm{OPT}$. As above, with probability at least $1 - \beta/3$, all Laplace noises sampled in Line 9 are bounded by $\frac{6\Delta_f}{\varepsilon} \log \frac{3n}{\beta}$, and the candidate set considered in Line 11 includes OPT. Conditioning on this event, the output $S^*$ therefore satisfies

$$f(S^*) \geq f(\mathrm{OPT}) - \frac{12\Delta_f}{\varepsilon} \log \frac{3n}{\beta},$$

which completes the proof. □

### B.1.3. IMPROVED COMPLEXITY DEPENDENCE ON $\beta$

In Section 4.1, we noted that the query complexity in Theorem 4.1 can be reduced to $O(nk^3 \log(1/\beta))$ at the cost of an additional $\log(1/\beta)$ factor in the additive error. We provide a proof sketch of this modification below

Recall that in the standard configuration of Algorithm 2, the threshold $p \in [0, 1]$ is sampled uniformly, and each partial enumeration step is repeated $\lceil 3/\beta \rceil$ times. This corresponds to an instantiation of the selection mechanism $\texttt{Selection}_{\tau, \gamma}$ with $\tau = \lceil 3/\beta \rceil$ and $\gamma = 1$, applied to the collection of mechanisms $\{\mathcal{M}_Y \mid Y \subseteq \mathcal{N}, |Y| \leq 2, c(Y) \leq B\}$. To achieve the improved query complexity, we instead sample $p \in [0, 1]$ according to the cumulative distribution function $\mathbf{Pr}[p \leq x] = x^{\log(6/\beta)}$, corresponding to $\gamma = \log(6/\beta)$, and set the repetition threshold to $\tau = \lceil e/\log(6/\beta) \rceil$. This leads to a query complexity of $O(nk^3 \log(1/\beta))$.

To analyze the resulting utility, observe that $\mathbf{Pr}[p \leq 1/e] = (1/e)^{\log(6/\beta)} = \beta/6$. Thus, $p \geq 1/e$ holds with probability at least $1 - \beta/6$. Conditioned on this event, the probability that $\mathcal{M}_{Y^*}$ is executed at least once across $\tau$ trials is at least

$$1 - (1 - 1/e)^\tau \geq 1 - \exp(-\tau/e) \geq 1 - \beta/6.$$

By a union bound, $\mathcal{M}_{Y^*}$ is executed at least once with probability at least $1 - \beta/3$, allowing the downstream utility analysis of Lemma 5.6 to proceed unchanged. Finally, by Theorem A.5, each invocation of Algorithm 1 now needs to use the tighter privacy parameter $\varepsilon_0/(2(2 + \gamma)) = O(\varepsilon_0/\log(1/\beta))$ instead of $\varepsilon_0/6$ that was used in the proof of Lemma B.2 (which corresponds to $\gamma = 1$). Propagating this scaled parameter leads directly to the additional $O(\log(1/\beta))$ factor in the final additive error bound.

## B.2. Proof of Theorem 4.2

We now present a more efficient DP algorithm for the monotone SMK problem. Algorithm 7 is based on the Greedy$^+$ algorithm of (Yaroslavtsev et al., 2020), which executes density-greedy with the objective $f$ and returns the best solution among the partial solutions and all their feasible single-element extensions. Algorithm 7 executes Algorithm 1, and uses the RNM mechanism to select the best solution among the partial solutions, their feasible single-element extensions, and all singletons. This yields the guarantee stated in Theorem 4.2, which is proven by combining Lemmas B.8 to B.10.

---

**Algorithm 7** DP-DG$^+$ (DP-Density-Greedy$^+$)

---

**Input:** Dataset $D$; Submodular function $f_D : 2^{\mathcal{N}} \to \mathbb{R}_+$; Budget $B$; Parameters $\beta, \varepsilon, \delta \in (0, 1]$.

1  $S_0 \leftarrow \varnothing$ and $i \leftarrow 0$.

2  $\varepsilon_0 \leftarrow \frac{\varepsilon}{4\sqrt{2k \log(1/\delta)}}$

3  Let $S_1, \ldots, S_\ell$ be all the partial solutions obtained throughout the execution of Algorithm 1 on the instance defined by the ground set $\mathcal{N}$, the objective function $h_D(S) = f_D(S)$, the budget $B$, the privacy parameter $\varepsilon_0$, and utility parameter $\beta/2$.

4  Define $\mathcal{S} \leftarrow \{S_\ell\} \cup \{\{u\} \mid u \in \mathcal{N}\} \cup \{S_i \cup \{u\} \mid 0 \leq i \leq \ell, \ u \in \mathcal{N}, \ c(S_i \cup \{u\}) \leq B\}$,

5  Let $S^* \leftarrow \mathtt{RNM}_{\varepsilon/2}(\mathcal{S}, f; D)$

6  **return** $S^*$

---

**Lemma B.8.** *Algorithm 7 performs $O(nk)$ oracle queries.*

*Proof.* Algorithm 1 executes at most $k$ iterations of its main loop, and in each iteration, it evaluates the marginal gain of at most $n$ elements, contributing $O(nk)$ oracle queries. The final invocation of RNM in Line 5 requires one oracle call for each candidate in the set $\mathcal{S}$, which contains at most $nk + n + 1$ candidates. Consequently, the total number of oracle queries made by Algorithm 7 is $O(nk)$. $\qquad\square$

**Lemma B.9.** *For any $\varepsilon, \delta \in (0, 1]$, Algorithm 7 is $(\varepsilon, \delta)$-DP.*

*Proof.* The privacy guarantee follows from the composition of the iterative greedy phase and the final selection phase. First, we consider the execution of Algorithm 1 in Line 3. By instantiating Algorithm 1 with the per-step privacy parameter $\varepsilon_0 = \varepsilon/(4\sqrt{2k \log(1/\delta)})$, advanced composition (Theorem 3.2) implies that releasing the sequence of partial solutions $S_1, \ldots, S_\ell$ is $(\varepsilon/2, \delta)$-DP. Second, the final selection step via RNM is executed with privacy parameter $\varepsilon/2$. According to Theorem A.2, this step is $(\varepsilon/2, 0)$-DP. Combining this with the privacy guarantee of Algorithm 1 via basic composition yields that the overall algorithm is $(\varepsilon, \delta)$-DP. $\qquad\square$

**Lemma B.10.** *For any $\varepsilon, \delta > 0$, $\beta \in (0, 1)$, Algorithm 7 returns a feasible subset $S$ such that, with probability $1 - \beta$,*

$$f(S) \geq \tfrac{1}{2} \cdot f(\mathrm{OPT}) - O\left(\frac{\Delta_f \, k^{1.5}}{\varepsilon} \sqrt{\log \tfrac{1}{\delta}} \log \tfrac{n}{\beta}\right).$$

*Proof.* We consider two cases based on the cardinality of OPT and the objective value of each single-element extension of a partial solution. Let $o_m \in \arg\max_{o \in \mathrm{OPT}} c(o)$ and let $T$ be the smallest index in $\{1, \ldots, \ell-1\}$ satisfying $c(S_T) > 1 - c(o_m)$, or let $T = \ell$ if no such index exists.

**Case 1.** If $|\mathrm{OPT}| = 1$, then $\mathrm{OPT} \in \mathcal{S}$ as all singletons are included in the candidate set. If for some $0 \leq i < T$, $f(S_i \cup \{o_m\}) \geq \frac{1}{2} f(\mathrm{OPT})$, then there exists a set $S' \in \mathcal{S}$ such that $f(S') \geq \frac{1}{2} f(\mathrm{OPT})$. In both scenarios, the utility guarantee of RNM (Theorem 3.5) ensures that with probability at least $1 - \beta$:

$$f(S^*) \geq \max_{S \in \mathcal{S}} f(S) - \frac{4\Delta_f}{\varepsilon} \log \frac{nk + n + 1}{\beta} \geq \frac{1}{2} f(\mathrm{OPT}) - O\left(\frac{\Delta_f}{\varepsilon} \log \frac{n}{\beta}\right).$$

**Case 2.** We now consider the case where $|\mathrm{OPT}| \geq 2$ and $f(S_i \cup \{o_m\}) < \frac{1}{2} f(\mathrm{OPT})$ for all $i < T$. Following a chain of inequalities analogous to that in the proof of Lemma B.4, we obtain that with probability at least $1 - \beta/2$,

$$(1 - c(o_m)) \cdot \frac{f(u_{i+1} \mid S_i)}{c(u_{i+1})} \geq f(\mathrm{OPT}) - f(S_i \cup \{o_m\}) - kE$$

$$\geq \tfrac{1}{2} f(\mathrm{OPT}) - kE, \qquad\qquad \forall i \in \{0, \ldots, T-1\},$$

where $E = \frac{8\Delta_f}{\varepsilon_0} \log \frac{2nk}{\beta}$. Rearranging the inequality yields:

$$\frac{c(u_{i+1})}{1 - c(o_m)} \cdot \left(\frac{f(\mathrm{OPT})}{2} - kE\right) \leq f(S_{i+1}) - f(S_i).$$

We may assume $f(\text{OPT}) \geq 2kE$, as otherwise the guarantee holds trivially. Summing over $0 \leq i \leq T-1$ yields:

$$f(S_\ell) \geq f(S_T) \geq f(S_0) + \left(\frac{f(\text{OPT})}{2} - kE\right) \cdot \sum_{i=0}^{T-1} \frac{c(u_{i+1})}{1 - c(o_m)}$$

$$\geq \left(\frac{f(\text{OPT})}{2} - kE\right) \cdot \frac{c(S_T)}{1 - c(o_m)} \geq \frac{f(\text{OPT})}{2} - kE, \tag{5}$$

where we use the identity $\sum_{i=0}^{T-1} c(u_{i+1}) = c(S_T)$ and the non-negativity of $f$. It therefore remains to show that $c(S_T) \geq 1 - c(o_m)$. If $T < \ell$, this inequality holds by the definition of $T$. Thus, assume that $T = \ell$. By monotonicity of $f$, it suffices the consider the case that $\text{OPT} \not\subseteq S_\ell$. In this case, there exists an element $o \in \text{OPT} \setminus S_\ell$. Since Algorithm 1 has terminated and $o$ was not added to the solution, it must be that $c(S_\ell) > 1 - c(o) \geq 1 - c(o_m)$.

Observe that since $S_\ell \in \mathcal{S}$, Theorem 3.5 implies that with probability at least $1 - \beta/2$:

$$f(S^*) \geq f(S_\ell) - \frac{4\Delta_f}{\varepsilon_0} \log \frac{nk + n + 1}{\beta/2}.$$

By a union bound and substituting $\varepsilon_0 = \frac{\varepsilon}{4\sqrt{2k \log(1/\delta)}}$, we conclude that with probability at least $1 - \beta$:

$$f(S^*) \geq \tfrac{1}{2} f(\text{OPT}) - O\left(\frac{\Delta_f \, k^{1.5}}{\varepsilon} \sqrt{\log \tfrac{1}{\delta}} \, \log \tfrac{n}{\beta}\right).$$

$\square$

## C. Proofs from Section 6

In this section, we provide the proofs deferred from Section 6 on non-monotone submodular maximization, completing the proof of Theorem 4.3. For convenience in the utility analysis, we adopt a standard analytical setup where the DP noise random variables are realized at the beginning of execution. See Section C.2 for the formal treatment and definitions.

The query complexity of Algorithm 3 is given by the following lemma.

**Lemma C.1.** *Algorithm 3 makes $O(nk)$ oracle queries.*

*Proof.* Let $S_\ell = \{u_1, \ldots, u_{|S_\ell|}\}$ be the solution obtained at the end of the while loop. Algorithm 3 can be implemented to compute $O(n)$ marginal gains for each prefix $\{u_1, \ldots, u_j\}$. Because $S_\ell$ is feasible, $|S_\ell| \leq k$. Thus, the total number of oracle queries incurred within the while loop is $O(nk)$. Finally, the last application of RNM in Algorithm 3 evaluates the objective function on each set in $\mathcal{S}$, defined in Line 11. Since we have at most $k$ distinct partial solutions, the candidate family $\mathcal{S}$ contains at most $k + k \cdot n = O(nk)$ sets. Hence this step also requires $O(nk)$ oracle queries. Overall, the total query complexity is $O(nk)$. $\square$

### C.1. Privacy Analysis

We now proceed to the privacy analysis. To this end, we utilize the following concentration bound for the sum of independent geometric random variables.

**Lemma C.2** (Janson, 2018)**.** *Let $X_1, \ldots, X_k$ be independent, geometrically distributed, random variables with $X_i \sim \text{Geom}(p_i)$ for $0 < p_i \leq 1$. Let $p_* = \min_i p_i$, and $\mu = \mathbb{E}[\sum_{i=1}^{k} X_i]$. Then, for every $\lambda \geq 1$,*

$$\mathbf{Pr}\left[\sum_{j=1}^{k} X_j \geq \lambda\mu\right] \leq \exp\left(-p_*\mu\left(\lambda - 1 - \log\lambda\right)\right).$$

We also rely on the following technical inequality.

**Lemma C.3.** *For all $x > 0$, we have*

$$x - \log(1+x) \geq \frac{x^2}{2(1+x)}.$$

*Proof.* Define $g(x) := x - \log(1+x) - \frac{x^2}{2(1+x)}$. Then

$$g'(x) = 1 - \frac{1}{1+x} - \frac{x(2+x)}{2(1+x)^2} = \frac{x^2}{2(1+x)^2} \geq 0.$$

Since $g(0) = 0$ and $g$ is non-decreasing for $x > 0$, we conclude $g(x) \geq 0$ for all $x > 0$. $\qquad\square$

As a consequence, we obtain the following lemma.

**Lemma C.4.** *Suppose $X_i \sim \mathrm{Geom}(1/2)$ for all $i$. Then, for every $\delta > 0$,*

$$\mathbf{Pr}\left[\sum_{j=1}^{k} X_j \geq (2+\sqrt{2})k + (4+\sqrt{2})\log(1/\delta)\right] \leq \delta.$$

*Proof.* Let $X = \sum_{i=1}^{k} X_i$, and note that we have $p_* = 1/2$ and $\mu = \mathbb{E}[X] = 2k$. For a parameter $t > 0$ to be specified later, let $\lambda = 1 + \frac{t}{2k}$. Then by Theorem C.2,

$$\mathbf{Pr}[X \geq 2k + t] \leq \exp\left(-k\left(\frac{t}{2k} - \log\left(1 + \frac{t}{2k}\right)\right)\right)$$

$$\leq \exp\left(-k \cdot \frac{(t/2k)^2}{2(1 + t/2k)}\right) = \exp\left(-\frac{t^2}{4(2k+t)}\right) \tag{6}$$

where the second inequality uses Lemma C.3, applied with $x = \frac{t}{2k}$.

In order for the above probability to be at most $\delta$, it suffices that

$$\frac{t^2}{4(2k+t)} \geq \log(1/\delta)$$

$$\Longleftrightarrow \quad t^2 - 4t\log(1/\delta) - 8k\log(1/\delta) \geq 0.$$

Solving the quadratic inequality, we find that whenever

$$t > 2\log(1/\delta) + \sqrt{4\log^2(1/\delta) + 8k\log(1/\delta)}$$

By (6) we have $\mathbf{Pr}[X \geq 2k + t] \leq \delta$. To further simplify the bound and complete the proof, observe that

$$\sqrt{2}k + (4+\sqrt{2})\log(1/\delta) \;\geq\; 4\log(1/\delta) + 2\sqrt{2k\log(1/\delta)}$$

$$= \; 2\log(1/\delta) + 2\log(1/\delta) + 2\sqrt{2k\log(1/\delta)}$$

$$\geq \; 2\log(1/\delta) + 2\log(1/\delta)\left(1 + \sqrt{\tfrac{2k}{\log(1/\delta)}}\right)$$

$$\geq \; 2\log(1/\delta) + 2\log(1/\delta)\sqrt{1 + \tfrac{2k}{\log(1/\delta)}}$$

$$= \; 2\log(1/\delta) + \sqrt{4\log^2(1/\delta) + 8k\log(1/\delta)},$$

where the first inequality follows from the AM-GM inequality, which implies that $\sqrt{2}k + \sqrt{2}\log(1/\delta) \geq 2\sqrt{2k\log(1/\delta)}$. $\qquad\square$

**Lemma C.5.** *Algorithm 3 executes the* `while` *loop (Lines 3-10) at most $(2+\sqrt{2})k + (4+\sqrt{2})\log(2/\delta)$ times with probability at least $1 - \delta/2$.*

*Proof.* For the purpose of analysis, we assume without loss of generality that if the loop terminates with $|S_\ell| < k$, the algorithm sets $S_{\ell+1}, \ldots, S_k \leftarrow S_\ell$. This postprocessing does not affect the privacy analysis and ensures the algorithm executes exactly $k$ phases.

For $j = 1, \ldots, k$, let $N_j$ denote the number of iterations of the `while` loop during which the algorithm has selected exactly $j - 1$ elements. Formally, if $j \leq \ell$, then $N_j$ is the number of iterations from the addition of the $(j - 1)$-th element up to and including the addition of the $j$-th element. If $j > \ell$, we set $N_j = 0$. The algorithm remains in phase $j$ only if the random inclusion step in Line 8 rejects the selected element. Since the probability of acceptance in each trial is $1/2$ and independent of the prior state, the number of trials until an element is accepted follows a geometric distribution with parameter $1/2$. We couple the execution with a sequence of $k$ independent random variables $X_1, \ldots, X_k \sim \mathrm{Geom}(1/2)$ ensuring that $N_j \leq X_j$ for all $j$. For each phase $j \leq \ell$, we have $N_j = X_j$, representing the number of trials until an acceptance. For $j > \ell$, we have $N_j = 0 \leq X_j$. This coupling implies that the total number of iterations $\sum_{j=1}^{k} N_j$ is stochastically dominated by $\sum_{j=1}^{k} X_j$. That is, for all $t \geq 0$:

$$\mathbf{Pr}\left[\sum_{j=1}^{k} N_j > t\right] \leq \mathbf{Pr}\left[\sum_{j=1}^{k} X_j > t\right].$$

The lemma follows by applying Lemma C.4 to the right-hand side with $\delta' = \delta/2$. □

We now prove the privacy guarantee of Algorithm 3.

**Lemma C.6.** *For any $\varepsilon, \delta > 0$, Algorithm 3 is $(\varepsilon, \delta)$-DP.*

*Proof.* Define $L = (2 + \sqrt{2})k + (4 + \sqrt{2})\log(2/\delta)$. Each iteration of the `while` loop (Lines 3–10) applies the GRNM with privacy parameter $\varepsilon_0$, where

$$\varepsilon_0 = \frac{\varepsilon}{4\sqrt{2L\log(2/\delta)}}.$$

Furthermore, each quality score $q_i(u; D)$ has sensitivity $2\Delta_f / c(u)$, by an argument similar to that in the proof of Theorem B.3, and $f_D(u_{i+1} \mid S_i)$ has sensitivity $2\Delta_f$. Thus, by Theorem 3.5 each single iteration, which releases a candidate element $u_{i+1}$, is $\varepsilon_0$-DP. Advanced composition (Theorem 3.2) implies that releasing the outputs of the first $L$ iterations is $(\varepsilon/2, \delta/2)$-DP. By Lemma C.5, the loop exceeds $L$ iterations with probability at most $\delta/2$. A standard union bound over these two failure events implies that the algorithm up to Line 11 is $(\varepsilon/2, \delta)$-DP. Finally, by basic composition with the final RNM step (Line 12), which uses privacy parameter $\varepsilon/2$, the overall algorithm is $(\varepsilon, \delta)$-DP. □

### C.2. Utility Analysis

To simplify the analysis of Algorithm 3, we assume, without loss of generality, that all differential privacy noise random variables are sampled at the beginning of the algorithm. As noted in Remark A.4, for each application of GRNM within Algorithm 3, we use $n$ as a uniform upper bound for the candidate set size instead of the iteration-specific $|\mathcal{C}_i|$. Consequently, we set $t = \frac{2}{\varepsilon_0} \log \frac{2n^2}{\beta}$. This allows us to isolate the randomness of the sampling step in Line 8. Specifically, we assume the following random variables are sampled upfront:

- $n^2$ independent samples from $\mathrm{Exp}\left(\frac{\varepsilon_0}{2}\right)$ for the GRNM calls in Line 6.

- $nk + k$ independent samples from $\mathrm{Exp}\left(\frac{\varepsilon}{4}\right)$ for the RNM call in Line 12.

The proof of the utility guarantee for Algorithm 3 follows the proof outlined by Han et al. (2021). We extend their techniques to account for the additive errors introduced by the DP mechanisms. We begin by introducing events under which these errors are bounded.

- **Event $\mathcal{E}_1$:** All $n^2$ samples from $\mathrm{Exp}\left(\frac{\varepsilon_0}{2}\right)$ are bounded by $\frac{2}{\varepsilon_0} \log \frac{2n^2}{\beta}$

- **Event $\mathcal{E}_2$:** All $nk + k$ samples from $\mathrm{Exp}\left(\frac{\varepsilon}{4}\right)$ are bounded by $\frac{4\Delta_f}{\varepsilon} \log \frac{2k(n+1)}{\beta}$.

- **Event $\mathcal{E}$:** $\mathcal{E} = \mathcal{E}_1 \cap \mathcal{E}_2$.

**Lemma C.7.** *Conditioned on $\mathcal{E}$, the following hold.*

*(i) Each element $u_{i+1}$ selected in Line 6 of Algorithm 3 satisfies*

$$q_i(u_{i+1}; D) = \frac{f(u_{i+1} \mid S_i)}{c(u_{i+1})} \geq \max_{u \in \mathcal{C}_i}\left\{\frac{f(u \mid S_i)}{c(u)} - \frac{8\Delta_f}{c(u)\varepsilon_0} \log \frac{2n^2}{\beta}\right\}. \tag{7}$$

*(ii) Algorithm 3 outputs a set $S^*$ satisfying*

$$f(S^*) \geq \max_{S \in \mathcal{S}} f(S) - \frac{4\Delta_f}{\varepsilon} \log \frac{2k(n+1)}{\beta}.$$

*Proof.* We begin by proving *(i)*. In each iteration $i$, GRNM is invoked with privacy parameter $\varepsilon_0$. The quality function $q_i(u; D)$ has a sensitivity of $2\Delta_f/c(u)$ with respect to the dataset argument. Under event $\mathcal{E}$, the $n^2$ exponential noise samples utilized across all GRNM invocations are bounded by $\frac{2}{\varepsilon_0} \log \frac{2n^2}{\beta}$. Adapting the utility analysis of GRNM (Theorem A.3) to this noise bound, we obtain

$$q_i(u_{i+1}; D) \geq \max_{u \in \mathcal{C}_i} \left\{ q_i(u; D) - \frac{4\Delta_{q_i(u;\cdot)}}{\varepsilon_0} \log \frac{2n^2}{\beta} \right\} = \max_{u \in \mathcal{C}_i} \left\{ q_i(u; D) - \frac{8\Delta_f}{c(u)\varepsilon_0} \log \frac{2n^2}{\beta} \right\}.$$

For *(ii)*, we consider the final RNM step in Line 12. Under event $\mathcal{E}$, all $n + nk$ realized exponential noises are bounded by $\frac{4\Delta_f}{\varepsilon} \log \frac{2k(n+1)}{\beta}$. Because RNM selects the element with the maximum noisy score, by Theorem A.2, the true value of the selected element differs from the true maximum value by at most the bound of the noise. This yields the stated bound. $\square$

**Lemma C.8.** *The event $\mathcal{E}$ occurs with probability at least $1 - \beta$.*

*Proof.* The lemma follows from applying tail bounds to each noise distribution followed by a union bound.

- Event $\mathcal{E}_1$: For $X \sim \mathrm{Exp}(\frac{\varepsilon_0}{2})$, we have $\mathbf{Pr}[X > s] = e^{-\varepsilon_0 s/2}$. Setting $s = \frac{2}{\varepsilon_0} \log \frac{2n^2}{\beta}$ gives $\mathbf{Pr}[X > s] = \frac{\beta}{2n^2}$. By a union bound over $n^2$ samples, $\mathbf{Pr}[\mathcal{E}_1^c] \leq \beta/2$.

- Event $\mathcal{E}_2$: For $Z \sim \mathrm{Exp}(\frac{\varepsilon}{4\Delta_f})$, setting $s = \frac{4\Delta_f}{\varepsilon} \log \frac{2k(n+1)}{\beta}$ gives $\mathbf{Pr}[Z > s] = \frac{\beta}{2k(n+1)}$. A union bound over $k(n+1)$ samples yields $\mathbf{Pr}[\mathcal{E}_2^c] \leq \beta/2$.

Combining these via a union bound over the complements $\mathcal{E}_1^c$, and $\mathcal{E}_2^c$ gives $\mathbf{Pr}[\mathcal{E}] \geq 1 - \beta$. $\square$

We now move on to introduce notation extended from (Han et al., 2021). Let $\ell$ be the last iteration of the while loop in Algorithm 3, and let $u_1, \ldots, u_\ell$ be the elements found in Line 6. Define $\tau(u_j) = j$ for $1 \leq j \leq \ell$ and $\tau(v) = \infty$ for any $v \in V \setminus \{u_1, \ldots, u_\ell\}$. Throughout the proof, we denote $E \triangleq \frac{8\Delta_f}{\varepsilon_0} \log \frac{2n^2}{\beta}$ which will serve as a bound for the error terms. For any realization of the randomness of the algorithm, define

$$r = \min \left\{ i \mid \left( \max_{\substack{v \in \mathrm{OPT} \setminus S_i \\ c(S_i \cup \{v\}) \leq B}} f(v \mid S_i) \leq E \right) \vee (i = \ell) \right\}$$

$$T = \min \left\{ i \mid \left( 0 \leq i \leq r - 1 \wedge c(S_i) + c(u_{i+1}) > c(\mathrm{OPT} \setminus \{o_m\}) \right) \vee (i = r) \right\},$$

$$O_{\leq T} = \{ v \in \mathrm{OPT} \mid \tau(v) \leq T \},$$

$$O_{>T} = \{ v \in \mathrm{OPT} \mid \tau(v) > T \wedge f(v \mid S_T) > E \}.$$

Intuitively, $r$ is the smallest index in $\{0, \ldots, \ell\}$ such that no remaining element $v \in \mathrm{OPT} \setminus S_i$ whose addition remains feasible provides a marginal gain exceeding $E$, if such an index exists. Otherwise, $r = \ell$. Moreover, $T$ is the smallest index in $\{0, \ldots, r - 1\}$ such that adding $u_{T+1}$ into $S_T$ would make the cost exceed $c(\mathrm{OPT} \setminus \{o_m\})$, if such an index exists. Otherwise, $T = r$, and we have $c(S_r) \leq c(\mathrm{OPT} \setminus \{o_m\})$.

The following lemma from (Han et al., 2021), constructs a fractional allocation of the cost of late optimal items onto elements of the algorithm's partial solution, which allows the proof to upper-bound the total marginal contribution of those optimal items in terms of elements already selected by the algorithm. Note that our definition of $O_{>T}$ includes a stricter condition on the marginal gain than the one used in (Han et al., 2021). Thus, the cost inequality $c(O_{>T} \setminus \{o_m\}) < c([S_T \setminus (\mathrm{OPT} \setminus \{o_m\})] \cup \{u_{T+1}\})$ required for the existence of $\Psi$ remains satisfied.

**Lemma C.9** (Lemma 1 in [Han et al., 2021](#)). *For any realization of the randomness in Algorithm 3 such that if $T < \ell$, there exists a mapping $\Psi$ satisfying the following properties. For each $u \in O_{>T} \setminus \{o_m\}$, $\Psi(u)$ is a set of 2-tuples such that each tuple $(v, \lambda_v(u)) \in \Psi(u)$ satisfies $v \in [S_T \setminus (\text{OPT} \setminus \{o_m\})] \cup \{u_{T+1}\}$ and $0 < \lambda_v(u) \le \min\{c(u), c(v)\}$. Moreover, we have:*

$$\sum_{(v, \lambda_v(u)) \in \Psi(u)} \lambda_v(u) = c(u), \qquad\qquad \forall u \in O_{>T} \setminus \{o_m\},$$

$$\sum_{\substack{u \in O_{>T} \setminus \{o_m\} \\ (v, \lambda_v(u)) \in \Psi(u)}} \lambda_v(u) \le c(v), \qquad \forall v \in [S_T \setminus (\text{OPT} \setminus \{o_m\})] \cup \{u_{T+1}\}.$$

**Lemma C.10** (Formal version of Lemma 6.4). *For each $u \in \mathcal{N}$, let $X_u$ be an indicator for the event that $u \in O_{\le T} \setminus (S_T \cup \{o_m\})$ or $u \in S_T \setminus (\text{OPT} \setminus \{o_m\})$. Then, conditioned on $\mathcal{E}$,*

$$f(S_T \cup \text{OPT}) \le f(S_T \cup \{o_m\}) + \sum_{u \in \mathcal{N}} X_u f(u \mid S_u) + f(S^*) - f(S_T) + O\left(\frac{k\Delta_f}{\varepsilon_0} \log \frac{n}{\beta}\right),$$

*where for any $j \in \{1, \dots, \ell\}$, $S_{u_j}$ denotes $S_{j-1}$, i.e., the set to which $u_j$ is considered for addition. For any $u \notin \{u_1, \dots, u_\ell\}$, define $S_u = \varnothing$.*

*Proof.* By submodularity, we have

$$f(S_T \cup \text{OPT}) - f(S_T \cup \{o_m\}) \le \sum_{u \in \text{OPT} \setminus (S_T \cup \{o_m\})} f(u \mid S_T)$$

$$\le \sum_{u \in O_{>T} \setminus \{o_m\}} f(u \mid S_T) + \sum_{u \in O_{\le T} \setminus (S_T \cup \{o_m\})} f(u \mid S_T) + (k-1)E. \qquad (8)$$

where the last inequality uses the fact that for any $v \in \text{OPT} \setminus (O_{\le T} \cup O_{>T} \cup \{o_m\})$, it holds that $f(v \mid S_T) \le E$ by definition of $O_{>T}$. Since there are at most $k - 1$ such elements, their total contribution is upper-bounded by $(k-1)E$.

Now recall that since $T \le r$, for every $j \in \{1, \dots, T\}$, by definition of $r$, the candidate set $C_{j-1}$ includes a feasible element $v$ such that $f(v \mid S_{j-1}) \ge E$. Therefore, by Lemma C.7 and under the conditioning on $\mathcal{E}$,

$$f(u_j \mid S_{j-1}) \ge \max_{v \in \mathcal{C}_{j-1}} \left\{ \frac{f(v \mid S_{j-1})}{c(v)} - \frac{E}{c(v)} \right\} \ge 0.$$

Moreover, conditioned on $\mathcal{E}$, since $S_T$ is among the candidates given to RNM in Line 12, we have

$$f(S^*) \ge f(S_T) - \frac{4\Delta_f}{\varepsilon} \log \frac{2k(n+1)}{\beta} \ge f(S_T) - E,$$

where we have also used that $\varepsilon_0 \le \varepsilon$. Therefore,

$$0 \le \sum_{u \in S_T \setminus (\text{OPT} \setminus \{o_m\})} f(u \mid S_u) + f(S^*) - f(S_T) + E. \qquad (9)$$

We now consider two cases.

**Case 1:** $T = r$. In this case, it must be that $O_{>T} = \varnothing$. Indeed, suppose toward a contradiction that $O_{>T} \ne \varnothing$. If $T = r = \ell$, then $c(S_\ell) \le c(\text{OPT} \setminus \{o_m\})$. Since $O_{>T} \subseteq \text{OPT}$, there exists an element in $O_{>T}$ that can be added to $S_\ell$ without violating the budget constraint, contradicting termination at iteration $\ell$. Thus, assume $T = r < \ell$. By the definition of $T$, any $v \in O_{>T}$ can be added to $S_T$ without violating the budget constraint. However, the definition of $r$ implies that for each such element, $f(v \mid S_T) \le E$, which contradicts the defining property of $O_{>T}$.

By (8) and (9), we get

$$f(S_T \cup \text{OPT}) - f(S_T \cup \{o_m\}) \le \sum_{u \in O_{\le T} \setminus (S_T \cup \{o_m\})} f(u \mid S_u) + \sum_{u \in S_T \setminus (\text{OPT} \setminus \{o_m\})} f(u \mid S_u)$$

$$+ f(S^*) - f(S_T) + kE \qquad (10)$$

The inequality in the lemma follows.

**Case 2:** $T < r$. This case leverages the mapping defined in Theorem C.9 to bound the term $\sum_{u \in O_{>T} \setminus \{o_m\}} f(u \mid S_T)$. We have

$$
\begin{aligned}
\sum_{u \in O_{>T} \setminus \{o_m\}} f(u \mid S_T) &= \sum_{u \in O_{>T} \setminus \{o_m\}} \frac{f(u \mid S_T)}{c(u)} \cdot c(u) \\
&= \sum_{u \in O_{>T} \setminus \{o_m\}} \frac{f(u \mid S_T)}{c(u)} \sum_{(v, \lambda_v(u)) \in \Psi(u)} \lambda_v(u) \\
&\leq \sum_{u \in O_{>T} \setminus \{o_m\}} \sum_{(v, \lambda_v(u)) \in \Psi(u)} \left( \frac{f(v \mid S_v)}{c(v)} + \frac{E}{c(u)} \right) \cdot \lambda_v(u) \\
&= \sum_{u \in O_{>T} \setminus \{o_m\}} \left[ E + \sum_{(v, \lambda_v(u)) \in \Psi(u)} \left( \frac{f(v \mid S_v)}{c(v)} \right) \cdot \lambda_v(u) \right] \\
&\leq \sum_{v \in (S_T \setminus (\mathrm{OPT} \setminus \{o_m\})) \cup \{u_{T+1}\}} \frac{f(v \mid S_v)}{c(v)} \cdot c(v) + kE \\
&\leq \sum_{v \in S_T \setminus (\mathrm{OPT} \setminus \{o_m\})} f(v \mid S_v) + f(u_{T+1} \mid S_T) + \frac{8k\Delta_f}{\varepsilon_0} \log \frac{2n^2}{\beta}. \\
&\leq \sum_{v \in S_T \setminus (\mathrm{OPT} \setminus \{o_m\})} f(v \mid S_v) + f(S^*) - f(S_T) + \frac{4\Delta_f}{\varepsilon} \log \frac{2k(n+1)}{\beta} + \frac{8k\Delta_f}{\varepsilon_0} \log \frac{2n^2}{\beta}.
\end{aligned}
$$

For the first inequality, note that no element $u \in O_{>T} \setminus \{o_m\}$ is selected before iteration $T$, and adding such an element to any $S_i$ for $i < T$ preserves feasibility. Consequently, $u$ appears in the candidate set provided to GRNM at the iteration in which $v$ is selected. Therefore, by submodularity and the guarantee in Lemma C.7, we obtain

$$
\frac{f(u \mid S_T)}{c(u)} \leq \frac{f(u \mid S_v)}{c(u)} \leq \frac{f(v \mid S_v)}{c(v)} + \frac{E}{c(u)}.
$$

For the final inequality, note that $S_T \cup \{u_{T+1}\}$ is included among the candidates provided to RNM at Line 12. Under $\mathcal{E}_2$, we have

$$
f(u_{T+1} \mid S_T) = f(S_T \cup \{u_{T+1}\}) - f(S_T) \leq f(S^*) - f(S_T) + \frac{4\Delta_f}{\varepsilon} \log \frac{2k(n+1)}{\beta}.
$$

Substituting this bound into (8) and using that $\varepsilon_0 \leq \varepsilon$ completes the proof.

$\square$

The next lemma from (Han et al., 2021) shows that $\sum_{u \in \mathcal{N}} X_u \cdot f(u \mid S_u)$ has the same expected value as $f(S_T) - f(\varnothing)$, where the expectation is over the Bernoulli random variables that determine if a selected element is discarded. The proof extends to our setting by conditioning on any arbitrary realization of the DP random variables. Crucially, because our algorithm samples all DP noise in advance, conditioning on these realizations does not introduce correlations with the Bernoulli trials. This allows us to obtain the equality in expectation conditioned on $\mathcal{E}$. We provide a proof sketch for completeness. Henceforth, all expectations are implicitly conditioned on $\mathcal{E}$.

**Lemma C.11** (Lemma 3 in Han et al., 2021). *We have* $\mathbb{E}[f(S_T) - f(\varnothing)] = \mathbb{E}[\sum_{u \in \mathcal{N}} X_u \cdot f(u \mid S_u)]$, *where* $X_u$ *is defined in Lemma 6.4.*

*Proof Sketch.* Let us condition on a specific realization of the randomness of the DP mechanisms such that $\mathcal{E}$ occurs. By the law of total expectation, it suffices to show the equality in the lemma conditioned on any such realization. For each $u \in \mathcal{N}$, define a random variable $R_u$ by $R_u = f(u \mid S_u)$ if $u \in S_T$ and $R_u = 0$ otherwise. As $f(S_T) - f(\varnothing) = \sum_{u \in \mathcal{N}} R_u$, by linearity of expectation it suffices to show that $\mathbb{E}[R_u] = \mathbb{E}[X_u \cdot f(u \mid S_u)]$ holds for every $u \in \mathcal{N}$. Fix an element $u$ and condition further on all random choices up to the moment when $u$ is considered by the algorithm in Line 8, or on an

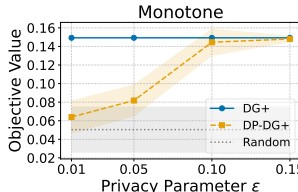

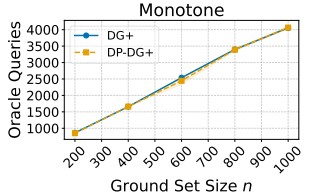

*(a)* Objective value for varying $\varepsilon$.  *(b)* Number of queries for varying $n$.

*Figure 3.* Performance of DP-DG$^+$ at an increased scale.

execution where the algorithm terminates and $u$ is never considered. Under this conditioning, the set $S_u$ is fixed. If $u$ is never considered, or if $c(S_u \cup \{u\}) > c(\text{OPT} \setminus \{o_m\})$ at the time it is considered, then $\tau(u) > T$, implying that both $R_u$ and $X_u$ are zero. Otherwise, we have $\tau(u) \leq T$. The algorithm adds $u$ into $S_u$ with probability $1/2$, and therefore $\mathbb{E}[R_u] = \frac{1}{2}f(u \mid S_u)$. We now consider the two cases in the definition of $X_u$. If $u \in \text{OPT} \setminus \{o_m\}$, then $X_u = 1$ if and only if $u$ is discarded by the algorithm. If $u \notin \text{OPT} \setminus \{o_m\}$, then $X_u = 1$ if and only if $u$ is selected by the algorithm. In both cases, $X_u = 1$ with probability $1/2$, and hence $\mathbb{E}[X_u \cdot f(u \mid S_u)] = \frac{1}{2}f(u \mid S_u)$. Since the conditional expectations agree in all cases, the claim follows. As the equality holds for any realization, it also holds in expectation conditioned on $\mathcal{E}$. $\square$

The proof of the utility guarantee is completed by utilizing the following lemma.

**Lemma C.12** (Buchbinder et al. (2014)). *Let $g : 2^{\mathcal{N}} \to \mathbb{R}$ be a non-negative submodular function. Denote by $Y$ a random subset of $\mathcal{N}$ where each element appears with probability at most $p$ (not necessarily independently). Then, $\mathbb{E}[g(Y)] \geq (1-p)g(\varnothing)$.*

**Lemma C.13** (Lemma 6.5 restated). *Algorithm 3 outputs a feasible set $S^*$ such that, with probability $1 - \beta$,*

$$\mathbb{E}[f(S^*)] \geq \frac{1}{4}f(\text{OPT}) - O\Big(\frac{k\Delta_f}{\varepsilon_0}\log\frac{n}{\beta}\Big),$$

*Proof.* The algorithm outputs a feasible set since all partial solutions considered are feasible, and only feasible sets are included in $\mathcal{S}$ as defined in Line 11. By Lemma C.8, we have $\mathbf{Pr}[\mathcal{E}] \geq 1 - \beta$. For the remainder of the proof, we condition on $\mathcal{E}$, and all expectations are implicitly conditioned on this event. Combining Theorem C.10 and Lemma C.11, we have:

$$\mathbb{E}[f(S_T \cup \text{OPT})] \leq \mathbb{E}[f(S_T \cup \{o_m\})] + \mathbb{E}[f(S_T) - f(\varnothing)] + \mathbb{E}[f(S^*) - f(S_T)] + O\Big(\frac{k\Delta_f}{\varepsilon_0}\log\frac{n}{\beta}\Big)$$

$$\leq 2\mathbb{E}[f(S^*)] + O\Big(\frac{k\Delta_f}{\varepsilon_0}\log\frac{n}{\beta}\Big).$$

The first inequality is by Theorem C.10 and Lemma C.11. The second inequality follows from non-negativity and the fact $S_T \cup \{o_m\}$ is among the candidates provided to RNM in Line 12. Therefore, conditioned on $\mathcal{E}$, we have $f(S^*) \geq f(S_T \cup \{o_m\}) - O(\frac{\Delta_f}{\varepsilon}\log\frac{n}{\beta})$.

Each element of $\mathcal{N}$ is included in $S_T$ with probability at most $1/2$ due to the Bernoulli sampling in Line 8. Hence, applying Lemma C.12 to the submodular function $g(S) \triangleq f(S \cup \text{OPT})$, we get

$$\mathbb{E}[f(S_T \cup \text{OPT})] = \mathbb{E}[g(S_T)] \geq \frac{1}{2}g(\varnothing) = \frac{1}{2}f(\text{OPT}).$$

Combining these inequalities, we obtain $\frac{1}{2}f(\text{OPT}) \leq 2\mathbb{E}[f(S^*)] + O(\frac{k\Delta_f}{\varepsilon_0}\log\frac{n}{\beta})$. Rearranging completes the proof. $\square$

## D. Additional Empirical Results

In this section, we present additional empirical results. We evaluate DP-DG$^+$ with a default setting of $n = 100, B = 20$ ($k = 8$), and scale to larger values of $n$, mirroring our analysis of DP-SDG in Section 7. We also study the impact of the knapsack capacity parameter $B$ on the utility of all algorithms.

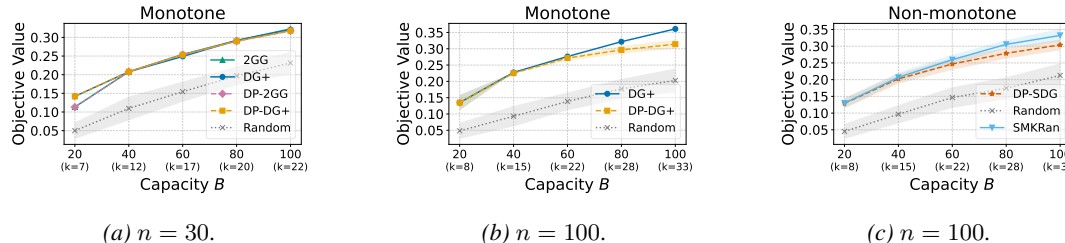

*(a)* $n = 30$.      *(b)* $n = 100$.      *(c)* $n = 100$.

*Figure 4.* Objective value for varying budget $B$. The induced maximal cardinality $k$ of a feasible solution is shown below the $x$-axis.

**Higher-Scale Evaluations for `DP-DG`$^+$.** The results in Figure 3a demonstrate that `DP-DG`$^+$ maintains strong utility relative to the non-private `DG`$^+$ baseline, achieving nearly identical performance even under a strict setting of $\varepsilon = 0.1$, where it remains within 3% of the baseline, while `Random` is 67% lower. Furthermore, Figure 3b highlights the algorithm's scalability: the number of queries grows linearly with $n$, consistent with the theoretical bound.

**Impact of $B$ on Utility.** Figure 4 shows the objective value obtained under different settings for varying values of $B$. As predicted by our theoretical error bounds, increasing $B$ leads to a decrease in the relative utility of the DP algorithms, with this degradation becoming most pronounced for the larger ground set size $n = 100$. Nevertheless, our proposed algorithms consistently maintain competitive utility across all regimes. As shown in Figure 4b for `DP-DG`$^+$ and in Figure 4c for `DP-SDG`, at $n = 100$ and $B = 80$ (corresponding to $k = 28$), both algorithms remain within 8% of their respective non-private counterparts, `DG`$^+$ and `SmkRan`, while `Random` performs substantially worse, achieving objective values at least 42% below the non-private baselines. Furthermore, Figure 4a shows that, in the monotone experiments with $n = 30$ (corresponding to $k = 22$), `DP-2GG` and `DP-DG`$^+$ remain within 1% of the non-private `2GG`.

