# OpenReview forum: "Differentially Private Submodular Maximization with a Knapsack Constraint"
_ICML.cc/2026/Conference — ICML 2026 regular_

### Official Review · Reviewer_Z1YL · 2026-03-04

**Soundness:** 3
**Presentation:** 3
**Significance:** 4
**Originality:** 4
**Overall Recommendation:** 5
**Confidence:** 4

**Summary:**

This paper is about differentially private submodular maximization under knapsack constraints, where private data defines the utility function, and the universe is public. dp submodular maximization has been studied with other constraint types, and only one other paper has studied this problem under knapsack constraints, only under monotone functions. This paper improves their result, and provides an algorithm for the non-monotone regime as well. In particular, while achieving the best multiplicative error that the non-private setting achieves, they reduce the additive error from depending linearly on n, the size of the ground set, to poly-logarithmically depending on n. they also present a more efficient algorithm with multiplicative approx 1/2 (all the algorithms are in polynomial time).

**Compliance With Llm Reviewing Policy:**

Affirmed.

**Final Justification:**

Thanks, will keep the score.

**Key Questions For Authors:**

Are there no lower bounds in any regime for dp submodular maximization?

**Limitations:**

Yes, They provide limitations in the form of future work.

**Strengths And Weaknesses:**

This paper is about a fundamental problem, and they use GRNM in a nice way which could be used in other problem, their algorithms are simple and practical.

There are a few minor presentation problems: define n when you mention N for the first time (|N|=n). Theorem A.3 and 3.5 must be the same thing but use different constants (2 vs 4).

---

> ### Author Rebuttal · Authors · 2026-03-30
>
> We are grateful for the reviewer’s positive evaluation of our work, and for highlighting the presentation issues. These will be addressed in the revision.
>
>
> **Q1 Are there no lower bounds in any regime for dp submodular maximization?**
>
> A1: We thank the reviewer for the question, and will include a dedicated discussion of lower bounds in the revised paper. Gupta et al. [6] proved a lower bound of $\Omega(k\log(n/k)/\epsilon)$ on the additive error of any $\epsilon$-DP algorithm for submodular maximization under a cardinality constraint. Chaturvedi et al. [7] extended this to the $(\epsilon, \delta)$-DP setting, proving that any algorithm achieving a multiplicative approximation factor $c$ must incur an additive error of $\Omega(k c \log(\epsilon/\delta)/\epsilon)$, assuming $n \ge k(e^\epsilon - 1)/\delta$ and $c \ge 4\delta/(e^\epsilon - 1)$. Since cardinality constraints are a special case of knapsack constraints, these lower bounds hold in our setting. However, in both [6] and [7], the hard instances utilize $1$-decomposable objective functions. Indeed, these bounds were shown to be tight (up to polylogarithmic terms) when the objective is decomposable. In contrast, our work, along with prior work such as [8], considers the broader class of bounded sensitivity submodular functions. It remains an open question whether the lower bounds from [6, 7] are tight for this more general class. Currently, our additive error is larger than these known lower bounds by a factor of $\tilde{O}(\sqrt{k})$ (Here $\tilde{O}$ omits polylogarithmic factors in the problem parameters). We will clarify this gap and the distinction between decomposable and general bounded-sensitivity functions in the revision.
>
>
> [6] Gupta et al. (SODA 2010)
>
> [7] Chaturvedi et al. (ICML 2023)
>
> [8] Mitrovic et al. (ICML 2017)

---

> > ### Author Rebuttal · Reviewer_Z1YL · 2026-04-02
> >
> > Thanks, will keep the score.

---

### Official Review · Reviewer_QXNF · 2026-03-12

**Soundness:** 2
**Presentation:** 2
**Significance:** 3
**Originality:** 3
**Overall Recommendation:** 4
**Confidence:** 4

**Summary:**

This paper investigates Differentially Private Submodular Maximization with a Knapsack Constraint (SMK). The core problem addressed is: how to privately find high-quality solutions while satisfying budget/knapsack constraints when the submodular objective function depends on sensitive data. The paper notes that previous work on differentially private submodular optimization has primarily focused on cardinality constraints or pseudo-array constraints, with relatively little research on knapsack constraints. Existing approaches also suffer from high query complexity, limited applicability to monotonic functions, and poor additivity error tolerance.
The paper's main contributions are twofold.
First, for monotonic submodular functions, the authors propose a differential privacy algorithm claiming to achieve the optimal $(1-1/e)$ approximation ratio while improving upon existing methods in terms of additive error and query complexity. Additionally, a more efficient variant achieving $1/2$ approximation is presented.
Second, for non-monotonic submodular functions, the authors present what they describe as the first provably guaranteed differential privacy algorithm, achieving $1/4$ approximation in the expected sense, with additive error bounds comparable to those in the monotonic case. The paper's conclusion reiterates: their results improve utility and query complexity in the monotonic setting, while establishing the first provable guarantee in the non-monotonic setting.

**Compliance With Llm Reviewing Policy:**

Affirmed.

**Final Justification:**

Score adjusted after rebuttal / discussion.

**Key Questions For Authors:**

1，Could you supplement with the most basic empirical validation, even if only through small-scale synthetic experiments? The current version of the paper primarily focuses on theoretical results; it transitions directly from the conclusions to the impact statement and then to the references, without presenting any empirical demonstration of the algorithm's behavior. Meanwhile, one of the paper's core selling points is improved additive error and query complexity, yet it remains unclear whether these improvements are perceptible under limited scale and privacy budgets. If the authors could supplement even very small-scale experiments—such as comparing actual utility degradation, query counts, and differences from prior methods under various $n,k,\varepsilon$ settings—it would significantly enhance my assessment of the paper's completeness and persuasiveness. Conversely, if the authors explicitly state why such experiments are unnecessary within this framework, please provide clear justification.
2，Could the author elaborate further on the specific parameter ranges within which the error bound improvements in this paper are “non-trivial”? For instance, compared to prior work, do these improvements manifest not only in asymptotic order but also yield significant advantages within achievable ranges of $n, k, \varepsilon, \delta$?
3. Regarding the claim of “the first guaranteed DP algorithm” in non-monotone scenarios, could the authors further clarify: What aspects of the technical novelty in this paper cannot be directly derived from existing non-private/DP tools? I would like to distinguish more clearly between setting-level novelty and method-level novelty.

**Limitations:**

No.The conclusion section does indeed mention two open problems: first, whether the $k^{1.5}$ dependence in the error term is tight, and second, that the optimal private query complexity remains unsolved. This attitude is commendable, but I believe the discussion of “limitations” remains somewhat narrow, primarily focusing on the incomplete theoretical improvements. Since the authors themselves have pointed out that the error dependence and optimal query complexity remain unresolved, they should also supplement the discussion with: how well the current results explain practical scenarios at finite scales, and whether there might be an overly conservative worst-case gap.

**Strengths And Weaknesses:**

This paper addresses the problem of maximizing the submodular objective under differential privacy constraints, covering both monotonic and non-monotonic objective functions. The authors' core claims are: for monotonic scenarios, they present a DP algorithm achieving optimal $(1-1/e)$ approximation while improving additive error and query complexity; for non-monotonic scenarios, they introduce what they claim is the first DP algorithm with provable guarantees, achieving expected $1/4$ approximation. The paper explicitly identifies these as its primary contributions and compares them with prior work in Table 1.
Overall, this is a theory-oriented algorithmic paper: the problem is relevant, the claims are clear, and the chain of results is largely complete. However, its persuasiveness relies almost entirely on theoretical guarantees rather than being supported by both theory and empirical evidence.
1. Soundness
This paper is well-grounded at the problem modeling level. It investigates submodular objectives driven by sensitive data, performing privacy-preserving optimization under knapsack constraints. The paper also provides a natural example of medical feature selection, illustrating why “cost constraints + privacy constraints + submodular objectives” may co-occur.
Regarding **technical claims**, the authors' assertions are clear and systematic: For monotonic scenarios, they claim to provide a DP algorithm achieving optimal $(1-1/e)$ approximation, further offering a more efficient $1/2$ approximation version. For non-monotonic scenarios, they claim to present the first DP algorithm with provable guarantees achieving an expected $1/4$ approximation.
Regarding proof organization, the paper does not merely make verbal claims but provides a relatively complete technical chain:
The main text first presents theorems/lemmas and proof sketches, with appendices completing the complexity, privacy analysis, and utility analysis for monotone and non-monotone cases. For instance, the monotonic section explicitly states in the main text that Theorem 4.1 is derived from the combination of Lemmas 5.1, 5.2, and 5.6, with the omitted proof provided in Appendix B. The non-monotonic section similarly relies on the combination of Lemmas 6.1, 6.3, and 6.5, with the full proof in Appendix C.
The non-monotonic section further breaks down the challenges into three distinct points: composition bottlenecks, noisy thresholding, and varying sensitivity, demonstrating the authors' clear understanding of the technical obstacles.
However, the theoretical foundation remains somewhat underdeveloped. First, the theory appears more like a refined assembly and privacy adaptation of existing tools rather than introducing a particularly novel analytical paradigm. The paper itself repeatedly states that its methods are based on existing non-private algorithmic paradigms and established DP selection mechanisms: for instance, the monotone part is a DP adaptation of density-greedy + privatized Two-Guess-Greedy, while the non-monotone part is a private variant of SmkRan combined with generalized private selection. This indicates that its theoretical novelty leans more toward “combination and extension.” Second, and more critically: the paper itself acknowledges in its conclusions that whether the $k^{1.5}$ dependency in the error term is tight remains an open problem, and that achieving better private query complexity has yet to be resolved. Third, the paper contains no experiments whatsoever. While this does not necessarily undermine its “correctness,” it directly impacts whether its claims are sufficiently supported. This is particularly problematic given the authors' emphasis on improved additive error and query complexity, yet they provide no synthetic instances to demonstrate whether these improvements are observable, stable, or more practical than prior methods at real-world scales. Since the arguments are almost entirely theorem-driven, this gap is strikingly evident. Indeed, the main text structure reveals no experiments section whatsoever.
2. Presentation Style:
The introduction provides a comprehensive explanation of the problem motivation and includes an example of feature selection to illustrate why combining knapsack problems with privacy protection holds practical significance. The related work section is well-organized, clearly separating non-private monotone, non-private non-monotone, and DP submodular maximization approaches. Table 1 is a significant plus, effectively comparing monotonicity, speed, non-monotonicity, and key metrics like approximation ratio, additive error, and query complexity across prior work. This greatly aids reviewers in understanding the paper's positioning. Additionally, the authors reasonably structure Section 4 by first clarifying technical challenges before introducing algorithms and proofs. Particularly in the non-monotone section, breaking down difficulties into three points enhances readability.
However, shortcomings exist. While the main text includes proof sketches, many critical technical details still rely on appendices, which dilutes the conciseness of the “core insights” presented in the main body.However, the absence of even basic experiments reduces readability and makes the presentation feel less intuitive.
3. Significance:
This paper investigates Differential Privacy-preserving Submodular Maximization with Knapsack Constraints. This problem is far from trivial, as submodular maximization (SMK) is a highly classical discrete optimization challenge. The paper's introduction enumerates numerous relevant application scenarios, including feature selection, recommendation systems, influence maximization, information gathering, active learning, data subset selection, and interpretable rule learning. The authors' motivation is straightforward: when sensitive data is involved, these tasks necessitate balancing optimization quality with privacy protection. They explicitly note that existing DP work has primarily focused on cardinality or matroid settings, while results for the knapsack setting remain relatively scarce. Furthermore, existing methods suffer from limitations such as supporting only monotone objectives, high query complexity, and suboptimal additive error dependence. The paper does not confine itself to the monotone case but explicitly addresses both monotone and non-monotone objectives. Non-monotone submodular optimization is typically more challenging and, theoretically, more prone to remaining confined to non-private settings. However, its significance lies primarily in theoretical contributions rather than empirical or practical applications.
4. Originality:
From the paper's presentation, the authors' method does not invent an entirely new optimization paradigm from scratch. Instead, it builds upon existing non-private SMK algorithms and DP tools: the monotone component relates to the Two-Guess-Greedy/density-based approach; the non-monotone component adopts the paradigm of practical non-private methods, namely density-greedy + randomized sampling. At the DP level, it further integrates tools like generalized private selection, noisy thresholding, and composition analysis. Thus, it creatively assembles existing combinatorial optimization frameworks and DP mechanisms to address novel technical challenges in knapsack/private settings while eliminating restrictive assumptions from prior theoretical results. Specifically, the authors highlight several limitations of previous methods: support only for monotone scenarios; Poor dependence of additive error on ground set size and element costs; suboptimal query complexity. The paper's novel results address these limitations.
However, shortcomings remain, as the authors themselves acknowledge in the conclusion. Significant open questions persist, such as whether the $k^{1.5}$ dependence in the error term is tight and whether tighter private query complexity can be achieved under weaker approximation guarantees.

---

> ### Author Rebuttal · Authors · 2026-03-31
>
> We thank the reviewer for the constructive review and helpful suggestions.
>
> **Q1: empirical validation**
>
> A1: As requested, we added experiments demonstrating the practicality of our approach. The results show that our algorithms are efficient and achieve utility comparable to non-private baselines even in strict privacy regimes. A representative sample of the results is provided in response A1 to Ng5q.
>
> Importantly, [5] is of a theoretical nature and does not provide implementation or empirical evaluation. Deriving a practical implementation is non-trivial, as the method relies on continuous optimization via the multilinear extension and omits details such as DP gradient approximation; thus, a direct comparison is currently infeasible. Accordingly, our experiments demonstrate the practicality and quality of our approach.
>
> **Q2: parameter ranges within which the error improvements are non-trivial**
>
> A2: As [5] omits explicit constants, exact numerical ranges are hard to state. However, our guarantees provide improvements for large $n$ or $B/c_{min} \gg k$. Specifically, our error scales with $\log^{1.5} n$ (vs. $n \log n$) and $k$ (vs. $B/c_{min}$). For intuition, consider a Max-Coverage instance ($B=10$, $\epsilon=\beta=0.1$, $|D|$ individuals, $\delta=|D|^{-1.5}$) with 499 sets of cost 1 and one outlier with cost 0.001, yielding $k=10$. Interpreting the additive error as the dataset size needed for a meaningful guarantee, our bound requires $\sim 10^4$ individuals, whereas [5] requires $\sim 10^8$.
>
> **Q3: in non-monotone scenarios what aspects cannot be directly derived from existing tools**
>
> A3: The non-monotone case is particularly challenging as analyses often rely on selecting elements of positive marginal gain, which we cannot ensure under DP.  While we build on SMKRan and leverage existing DP mechanisms, their integration and extension to both the analysis and algorithm design is not immediate. Simply replacing greedy selection with privatized ones breaks the analysis of SMKRan, and requires significant extensions to handle the possibility of negative gains. For instance, their analysis bounds the gain $f(S_T \cup OPT) - f(S_T \cup o_m)$ using the best feasible single-element extension. Since non-negativity is not guaranteed under DP, we prove that bounding it with $f(S^*)$ (up to a additive error) suffices.
>
> The closest DP algorithm to ours is Subsample-Greedy (Mitrovic et al. 2017) for cardinality constraints. However, their analysis varies significantly. Allowing only uniform costs, they use the Exponential Mechanism, the primary tool in DP submodular maximization. Our work is the first to introduce GRNM for greedy selection in this context, enabling comparable additive error in a more general setting. While they subsample the ground set per iteration, we must select each greedy element from the entire ground set and then discard it w.p. 1/2. Unlike the non-private setting, initial subsampling is not equivalent to post-selection discarding as the entire GRNM candidate set impacts its output distribution. This creates a unique challenge for a tight privacy analysis. To this end, we derive a concentration bound for iteration count, deviating notably from prior work that relied mostly on standard composition. We will better reflect this novelty in the revision.
>
> **theoretical foundation somewhat underdeveloped**
>
> A4: To maintain readability, we deferred most proofs to the appendix, which may have influenced this impression. While our results leverage existing non-private paradigms and DP mechanisms, their integration requires non-trivial innovations in both algorithm design and technical analysis (e.g., A2 to MaNU). In the revision, we will move more technical insights into the main body to better highlight these contributions.
>
> **discussion of limitations**
>
> A5: Thank you for the remarks.
>
> * The remaining gap is smaller than the current version of the paper might suggest. Our error exceeds known lower bounds by $\tilde{O}(\sqrt{k})$. However, the lower bounds are not proven tight, as detailed in A1 to Z1YL.
>
> * Ours are currently the fastest known DP-algorithms for SMK. DP Two-Guess-Greedy matches state-of-the-art complexity for non-private $(1-1/e)$-approximation algorithms, and the $\beta^{-1}$ factor can be tightened as detailed in A5 to Ng5q. While non-private $O(n)$ $1/2$-approximations exist, adapting them to DP is non-trivial and beyond our current scope. Yet, our results provide a significant first step toward such adaptations.
>
> * New experiments demonstrate that DP Subsample-Density-Greedy and DP Density-Greedy+ are highly practical, aligning with their complexity analysis. DP Two-Guess-Greedy is also practical for smaller instances. All algorithms outperform their worst-case utility guarantees, suggesting that the theoretical bounds may be conservative for real-world scenarios. We will include a full interpretation of the results in the revision.
>
> [5] Sadeghi and Fazel (AISTATS 2021)

---

> > ### Author Rebuttal · Reviewer_QXNF · 2026-04-03
> >
> > I appreciate the empirical proof of concept and explanation, and will adjust my score accordingly.

---

### Official Review · Reviewer_MaNU · 2026-03-12

**Soundness:** 3
**Presentation:** 3
**Significance:** 3
**Originality:** 2
**Overall Recommendation:** 4
**Confidence:** 3

**Summary:**

This paper focuses on the designing algorithms for Differentially Private Monotone and Non-monotone Submodular maximization subject to a Knapsak constraint. For monotone sub modular maximization the authors provide a stronger query complexity and error bound, while achieving the same approximation guarantee as the state of the art. They also present the first result for the non-monotone case.

**Compliance With Llm Reviewing Policy:**

Affirmed.

**Final Justification:**

I have shared my evaluation of the paper below and I keep my score.

**Key Questions For Authors:**

Please explain the weaknesses discussed above.

**Limitations:**

yes

**Strengths And Weaknesses:**

Overall:

The paper is overall well written.

Submodular maximization is well motivated with a wide range of applications, but DP with Knapsak constraints is not in the same level of interest. The authors also do not motivate the problem in this paper.

The novelty and new techniques and ideas are in the level expected for this conference. The challenges and the solutions presented in this paper are common in this literature.

Monotone case:

The results make significant improvements over the state of the art mainly in query complexity.

The algorithm is still quite theoretical and can only be used for small instances.

Non-monotone case:

I found this result also interesting and not as straightforward as one might expect.

---

> ### Author Rebuttal · Authors · 2026-03-30
>
> We would like to thank the reviewer for the detailed review. We hope that our response below addresses all their concerns about the paper. If further clarification is needed, we will be happy to provide it.
>
> **Q1: The authors also do not motivate the problem in this paper.**
>
>  A1: We will refine the presentation in our revised paper to more prominently feature the extensive applications of submodular maximization under knapsack constraints, particularly in settings involving sensitive data.
> There is vast literature on SMK algorithms designed to address practical ML tasks. DP SMK is strongly motivated by real world scenarios where resources are inherently budgeted and data privacy is required. Notable examples include feature selection (e.g., [3] or Example 1), revenue maximization in viral marketing [4], and personalized data summarization [2].
> Many of these applications have non monotone objectives, which are ubiquitous in ML contexts, particularly due to diversification and regularization penalties. Additionally, our newly added experiments, presented in response A1 to Ng5q, explore another highly relevant setting of budgeted ride-share optimization.
>
> Yet, existing methods for SMK are not privacy preserving, and cannot be applied in sensitive settings. Our work is the first to provide practical DP algorithms for this highly studied problem.
>
> [2] Mirzasoleiman et al. (ICML 2016)
>
> [3] Krause et al. (JMLR 2008)
>
> [4] Amanatidis et al. (NeurIPS 2020)
>
> **Q2: The challenges and the solutions presented in this paper are common in this literature.**
>
> A2: While certain high level challenges, such as handling noisy greedy selections and potentially negative marginal gains, have also been encountered in prior work, the core challenges we address, as well as the techniques we develop to overcome them, are novel and have not been treated in the existing literature on DP submodular maximization.
>
> Within this literature, our work is the first to:
> (i) introduce a privacy-efficient partial enumeration algorithm for submodular maximization via generalized selection, which enables tight privacy accounting and avoids standard composition based analysis (Theorem 4.1);
> (ii) Leverage the GRNM mechanism for SMK and develop a subtle analysis that harnesses its fine-grained guarantees to accurately handle heterogeneous sensitivities, resulting in significantly improved utility (Theorems 4.1
> to 4.3); and
> (iii) derive a novel privacy analysis by coupling the number of iterations with a sum of geometric random variables, which exhibits strong concentration and enables tighter privacy guarantees compared to naive composition (Theorem 4.3).
> We emphasize that our work is the first to successfully apply generalized selection mechanisms in the context of DP submodular maximization, representing a methodological departure from prior approaches.
>
>
> **Q3: The results make significant improvements mainly in query complexity.**
>
> A3: Improving query complexity in the monotone setting indeed constitutes one of our primary contributions, yielding the first practical algorithms for this problem, but is not the only contribution of the paper. We highlight our improvement in additive error as another central contribution. Our bounds bridge a notable gap in prior literature, matching the best known error for simpler cardinality constraints up to polylogarithmic terms. Our error scales with $\log^{1.5} n$ rather than $n \log n$ in prior work. Second, our bound avoids any explicit dependence on $c_{min}$ (the minimal element cost), scaling instead with $k$ (the maximal size of a feasible solution). This makes our guarantee more robust to varying cost scales, as demonstrated by the numerical example in our response A2 to Reviewer QXNF.
>
> **Q4: The algorithm is still quite theoretical and can only be used for small instances.**
>
> A4: We acknowledge that our presentation may not have sufficiently highlighted the practical merits of DP-Density-Greedy+ (DP-DG+), which was specifically designed to address the scalability challenges of DP Two-Guess-Greedy. DP Two-Guess-Greedy matches the best known complexity in the non-private setting among algorithms achieving the tight $(1-1/e)$ approximation, but may still not be sufficiently scalable for large instances. To overcome this, as a practical alternative, we designed DP-DG+. DP-DG+ is significantly faster, obtaining $O(nk)$ query complexity while maintaining strong theoretical guarantees. Furthermore, to demonstrate its scalability, we have now performed experiments showing that it performs comparably in practice with a vastly reduced number of oracle queries. Please see response A1 to N5q for experimental results, which will be added to the revised paper.

---

> > ### Author Rebuttal · Reviewer_MaNU · 2026-04-01
> >
> > Thanks for the rebuttal.
> >
> > I agree with the points made by the authors and the challenges and techniques are more clear.
> >
> > The concern about the motivation and practicality of the works is still there, even after reading the rebuttal.
> >
> > I keep the score that I had.
> >
> > A note to the authors:
> >
> > It is perfectly acceptable to submit a paper without experiments, just as it is acceptable to provide supplementary experimental results during the rebuttal in response to reviewers' feedback. However, introducing an entirely new section for experiments at this stage is neither expected nor encouraged in my opinion. Such a substantial revision should be reserved for a resubmission, as incorporating it within the strict page limits would require restructuring the entire paper.

---

> > > ### Author Response · Authors · 2026-04-01
> > >
> > > Thank you for the response and for acknowledging the increased clarity of our technical contributions. We would like to clarify our approach for the final version:
> > >
> > > **We do not intend to perform a substantial revision or restructure the paper.**
> > > The original structure, focus, theoretical development, and core content will remain unchanged. We plan to use the one extra page in the final version to include a concise summary of the empirical results, which provide a demonstration of the theory without altering the paper’s focus.
> > >
> > > This allows us to provide the empirical validation requested by two reviewers without compromising the original content of the submission. We believe that this minimal validation, developed and executed during the rebuttal period, helps address concerns regarding practicality while preserving the paper’s identity as a primarily theoretical contribution.
> > >
> > > As these evaluations indicate that our algorithms are practical, we hope this additional evidence helps alleviate your concerns, and we would sincerely appreciate your consideration of this evidence in your final assessment.

---

### Official Review · Reviewer_Ng5q · 2026-03-16

**Soundness:** 3
**Presentation:** 3
**Significance:** 3
**Originality:** 3
**Overall Recommendation:** 5
**Confidence:** 3

**Summary:**

The paper gives the first polynomial-time differentially private algorithms
for monotone and non-monotone submodular maximization under a knapsack
constraint (SMK). The only prior work on DP + SMK (Sadeghi & Fazel, AISTATS
2021) operates on the multilinear extension with exact gradient evaluations,
which costs 2^|V| value oracle queries per step — effectively exponential
time. This paper replaces the continuous relaxation with purely combinatorial
algorithms, achieving polynomial query complexity while preserving the
optimal multiplicative approximation ratios.

Three results:
- Monotone: (1 - 1/e)-approximation, O(β⁻¹n³k) queries (DP Two-Guess Greedy)
- Monotone: 1/2-approximation, O(nk) queries (DP Density Greedy+)
- Non-monotone: 1/4-approximation, O(nk) queries (DP Subsample Density Greedy)

All are (ε, δ)-differentially private. Privacy costs only additive error;
the multiplicative ratios exactly match the best non-private combinatorial
algorithms for SMK.

**Compliance With Llm Reviewing Policy:**

Affirmed.

**Final Justification:**

Score adjusted after rebuttal / discussion.

**Key Questions For Authors:**

1. For what ranges of ε, n, and k does the additive error term remain
   small relative to the objective value? A back-of-envelope calculation
   or a simple experiment would help readers assess practical relevance.

2. Is there a known lower bound for the additive error dependence on k?
   If the k^{1.5} term is loose, what is the suspected tight dependence?

3. In the Two-Guess Greedy algorithm, the enumeration over all pairs of
   elements contributes an O(n²) factor. Have you considered heuristics
   (e.g., enumerating only high-value or high-density pairs) that would
   reduce this in practice while preserving the worst-case guarantee?

4. The GRNM mechanism adapts noise to each element's individual
   sensitivity 2Δ_f/c(u). For elements with very small costs (c(u) → 0),
   the sensitivity diverges. How does the algorithm handle this in
   practice? Is there an implicit assumption that β = c_min/c_max is
   bounded away from zero?

**Limitations:**

yes

**Strengths And Weaknesses:**

## Strengths

- The contribution is real and clearly framed. Making an exponential-time
  algorithm polynomial while preserving the optimal approximation ratio is
  a qualitative advance — not just filling a cell in a settings table.

- The paper is honest about the relationship to prior work. A footnote
  acknowledges that sampling-based approximation of the multilinear
  extension is possible in principle but "would require a separate analysis
  beyond that in (Sadeghi & Fazel, 2021)." This is the right way to
  characterize the gap: the prior analysis doesn't yield a polynomial-time
  algorithm, even though a polynomial-time version might be achievable with
  new analysis. The paper fills this gap with a fundamentally different
  (combinatorial) approach rather than patching the continuous one.

- The use of GRNM (Generalized Report Noisy Max, Raskhodnikova & Smith
  2016) for density-greedy with heterogeneous element costs is well-
  motivated and non-trivial. In cardinality-constrained settings, all
  elements have unit cost, so standard EM suffices. Under a knapsack
  constraint, density scores f(u|S)/c(u) have sensitivity 2Δ_f/c(u) that
  varies by element — GRNM adapts noise to each element's individual
  sensitivity rather than the worst case. This is the right tool for the
  job and a good application of an underused mechanism.

- The Cohen et al. (2023) generalized private selection framework is
  applied well to avoid a composition penalty proportional to the n²
  enumeration in the Two-Guess Greedy algorithm. Without this, the privacy
  budget would blow up.

- The first non-monotone DP result for SMK is a legitimate contribution.
  The concentration analysis via tail bounds on sums of independent
  Geometric(1/2) random variables (Janson 2018) to control the iteration
  count is new and enables tight advanced composition.

- The multiplicative approximation ratios are preserved exactly: (1-1/e),
  1/2, and 1/4 match the best non-private combinatorial algorithms. Privacy
  degrades only the additive error term — a clean separation.

## Weaknesses

### 1. No experiments

For a paper on differential privacy, experimental evaluation matters more
than usual. Theoretical ε guarantees can be practically meaningless —
Apple's well-known DP implementation achieved ε ≈ 43 per day, which
provides essentially no privacy. The paper should demonstrate:

(a) For what ranges of ε and n the algorithms produce solutions that are
    both meaningfully private and meaningfully good (i.e., the additive
    error doesn't swamp the objective value).

(b) A concrete application (e.g., private feature selection, private
    influence maximization, private data summarization) where the
    knapsack constraint is natural and the privacy-utility tradeoff is
    evaluated empirically.

(c) How the O(β⁻¹n³k) query complexity of the optimal-ratio algorithm
    compares to the O(nk) complexity of the 1/2-approximation in
    practice — is the (1-1/e) algorithm ever worth running, or is the
    simpler 1/2-approximation always preferable?

This is the paper's most significant weakness. The theory is clean, but
the practical relevance is entirely asserted.

### 2. The O(β⁻¹n³k) query complexity for the optimal ratio is high

The Two-Guess Greedy algorithm enumerates all pairs of elements (O(n²)),
runs density-greedy on each residual instance (O(nk) steps), then uses
private selection across all candidates. The total query complexity is
O(β⁻¹n³k), where β = c_min/c_max is the cost ratio. For instances with
heterogeneous costs (β ≪ 1), this becomes impractical.

The 1/2-approximation variant avoids this entirely at O(nk), which raises
the question: is the gap between (1-1/e) ≈ 0.632 and 1/2 = 0.5 ever
worth the cubic blowup in practice? The paper would benefit from
discussing when each algorithm is preferable.

### 3. The k^{1.5} dependence in the additive error is not known to be tight

The authors acknowledge this as an open question, which is honest. But
for the paper to be fully compelling, some evidence — either a matching
lower bound or an empirical demonstration that the bound is not loose —
would help. As stated, the additive error could potentially be improved
by a polynomial factor, which would change the practical picture.

### 4. Only knapsack constraints

The results do not extend to matroid constraints, where Rafiey & Yoshida
(ICML 2020) have a polynomial-time (sampled) continuous greedy approach.
The paper's combinatorial approach is specific to the density-greedy
structure, which relies on the knapsack's cost-density ordering. This is
not a criticism of the paper's scope — the knapsack case is the one where
prior work was exponential — but it limits the generality of the
techniques.

## Minor Issues

- The relationship between the three algorithms could be presented more
  clearly as a Pareto frontier: (1-1/e) at O(n³k) cost vs. 1/2 at O(nk)
  cost vs. 1/4 for non-monotone. A table or figure showing this tradeoff
  would help readers choose the right algorithm for their setting.

- The paper's title says "Submodular Maximization" without specifying
  monotone or non-monotone. Since all three results are presented, this
  is fine, but the abstract should make clearer that the (1-1/e) and 1/2
  results are for monotone functions only.

---

> ### Author Rebuttal · Authors · 2026-03-30
>
> We would like to thank the reviewer for the insightful comments and helpful suggestions.
> The remarks regarding clarity will be addressed in the revision.
>
> **Q1: a simple experiment would help assess practical relevance**
>
> A1: To address the reviewer’s request, we evaluated our approach on a ride-share optimization task, the most common benchmark in the related literature (Mitrovic et al. 2017). The revised paper will include the full results; due to space constraints, we report here a representative sample.
> The goal is to select waiting spots for idle drivers using a dataset $D$ of $10^5$ Uber pickup locations in Manhattan. The ground set is a grid of $n$ points, with costs proportional to their distance from Penn Station.
>
> ## [Link to Figures](https://anonymous.4open.science/r/DP-SMK-Figures-CED6)
>
> * In the Monotone Setting:
> The objective is $f_D(S) = \frac{1}{|D|} \sum_{a \in D} \max_{b \in S} c(a,b)$ where $c(a,b)$ is the similarity between points $a,b$. We evaluated DP-Two-Guess-Greedy (DP-2GG) and DP-Density-Greedy+ (DP-DG+) against their non-private counterparts and a random baseline (default: $\epsilon=0.5$, $\delta = 1/|D|^{1.5}$, $n=30$, $B=40$, $k=12$). The results indicate our algorithms maintain high utility even in strict privacy regimes. At $\epsilon=0.2$, DP-2GG performs within 2% of the non-private baseline and DP-DG+ within 8%, whereas the Random baseline falls 49% below. DP-DG+ is significantly faster; at $n=60$, DP-2GG required $6000\times$ more queries than DP-DG+. Finally, we verified that DP-DG+ scales effectively up to $n=1000$, maintaining practical computational cost and high utility.
>
> * In the Non-Monotone Setting:  We consider a linear combination of the monotone objective with the cut function $\sum_{b \in S}\sum_{a \in \mathcal{N} \setminus S}c(a,b)$, aiming to promote geographic diversity among selected locations. We compared DP-Subsample-Density-Greedy (DP-SDG) with the non-private SMKRan and a random baseline (default: $\epsilon=1$, $\delta = 1/|D|^{1.5}$, $n=100$, $B=20$, $k=8$). DP-SDG maintains strong utility and practical complexity, staying within 2% of SMKRan at $\epsilon=1$, while Random is 60% below.
>
> **Q2: known lower bound for the additive error**
>
> A2: Thank you for this question. We will include an extended discussion of known lower bounds in the revised paper. As detailed in our response  A1 to Z1YL, our additive error exceeds known bounds by a factor of $\tilde{O}(\sqrt{k})$. Yet, the lower bounds are not known to be tight in our setting.
>
> **Q3: Have you considered heuristics**
>
> A3: In this work, we mostly focused on establishing theoretical guarantees. DP-DG+ is developed to address practical scalability concerns. Unfortunately, we are unaware of heuristics that reduce complexity while maintaining the same guarantees. Crucially, privately identifying a subset of high value pairs would incur additional privacy loss, and it is not immediately clear how to incorporate such a step without degrading the overall guarantee.
>
>
> **Q4: For elements with very small costs, the sensitivity diverges**
>
> A4: While one of our core contributions is a theoretical analysis that avoids an error term depending on the minimal cost, we acknowledge that any practical implementation of a DP algorithm must operate within the limits of machine precision. Our current implementation serves as a prototype to validate the proposed approach. In our experiments, costs are naturally sufficiently bounded from zero, and we observed no numerical instability.
>
> **Q5: The $O(\beta^{-1}n^3k)$ complexity is high**
>
> A5: Thank you for raising this point. The dependence on the failure probability $\beta$ is a tunable trade-off: while we prioritized additive error by setting $\tau=1/\beta$  and $\gamma=1$ in Algorithm 6, choosing $\tau=\log(1/\beta)$ and $\gamma=\log(1/\beta)$ reduces query complexity to $O(n^3k \log(1/\beta))$ with only an additional $\log(1/\beta)$ factor in additive error. We note that $O(n^3k)$ matches the best known non-private complexity. We will include this remark and derivation in the revised paper.
>
> **Q6: Only knapsack constraints, generality of the techniques**
>
> A6: In this work, we prioritize combinatorial algorithms as they are typically more efficient and easier to implement than continuous alternatives. As noted in Remark 3.1 of  [Rafiey & Yoshida], their stated query complexity does not fully account for approximating the multilinear extension, which likely results in a higher polynomial; additionally, their additive error is significantly larger than ours.
>
> Techniques such as enumerations and density scores are often used in non-private settings that combine knapsack and matroid constraints [1,2]. While these settings are beyond our current scope, we believe our approach paves the way for such generalizations. We will include this as an interesting direction for future work.
>
> [1] Badanidiyuru and Vondrák (SODA 2014)
>
> [2] Mirzasoleiman et al. (ICML 2016)

---

> > ### Author Rebuttal · Reviewer_Ng5q · 2026-04-02
> >
> > I appreciate the empirical proof of concept, and will adjust my score accordingly.

---

### Decision · Program_Chairs · 2026-04-30

**Decision:**

Accept (regular)

**Comment:**

The paper studies submodular maximization subject to a knapsack constraint with differential privacy. The previous works have studied the problem under other constraints like cardinality and matroid constraints. There are several technical challenges for this case and the paper overcomes them using fairly standard but sophisticated tools in the area. The paper has results for several settings including a more expensive but optimal 1-1/e approximation, more efficient algorithms with 1/2 approximation and an algorithm for the non-monotone case with 1/4 approximation. All reviewers are supportive of the paper for successfully complete an interesting gap in the literature. Thus, I recommend accepting the paper.

A minor comment: when talking about exponential time prior works, a baseline worth mentioning is to use the exponential mechanism over all possible solutions. This is already mentioned as a baseline for most problems in the work
Anupam Gupta, Katrina Ligett, Frank McSherry, Aaron Roth, Kunal Talwar. Differentially Private Combinatorial Optimization in SODA 2010.
This should get multiplicative factor 1, smaller additive error, and runtime roughly n^{k+O(1)}.